# IMPACT: Irregular Multi-Patch Adversarial Composition Based on Two-Phase Optimization

**Zenghui Yang[1,5], Xingquan Zuo[2,5]***, **Hai Huang[2,5], Gang Chen[3],**
**Xinchao Zhao[4], Tianle Zhang[1,5]**
[1]Shool of Cyberspace Security, Beijing University of Posts and Telecommunications
[2]School of Computer Science, Beijing University of Posts and Telecommunications
[3]School of Engineering and Computer Science, Victoria University of Wellington
[4]School of Science, Beijing University of Posts and Telecommunications
[5]Key Laboratory of Trustworthy Distributed Computing and Service
{yangzh,zuoxq,hhuang,zhaoxc,tlezhang}@bupt.edu.cn
aaron.chen@ecs.vuw.ac.nz

## Abstract

Deep neural networks have become foundational in various applications but remain vulnerable to adversarial patch attacks. Crafting effective adversarial patches is inherently challenging due to the combinatorial complexity involved in jointly optimizing critical factors such as patch shape, location, number, and content. Existing approaches often simplify this optimization by addressing each factor independently, which limits their effectiveness. To tackle this significant challenge, we introduce a novel and flexible adversarial attack framework termed IMPACT (Irregular Multi-Patch Adversarial Composition based on Two-phase optimization). IMPACT uniquely enables comprehensive optimization of all essential patch factors using gradient-free methods. Specifically, we propose a novel dimensionality reduction encoding scheme that substantially lowers computational complexity while preserving expressive power. Leveraging this encoding, we further develop a two-phase optimization framework: phase 1 employs differential evolution for joint optimization of patch mask and content, while phase 2 refines patch content using an evolutionary strategy for enhanced precision. Additionally, we introduce a new aggregation algorithm explicitly designed to produce contiguous, irregular patches by merging localized regions, ensuring physical applicability. Extensive experiments demonstrate that our method significantly outperforms several state-of-the-art approaches, highlighting the critical benefit of jointly optimizing all patch factors in adversarial patch attacks. Our source code is available at https://yangzh216.github.io/IMPACT.

## 1 Introduction

Deep Neural Networks (DNNs) have become one of the core technologies in modern artificial intelligence. With their exceptional learning ability, DNNs have demonstrated outstanding performance in fields such as image classification [21], object detection [3], and natural language processing [34], driving the rapid advancement of numerous real-world applications. However, recent studies demonstrate a concerning vulnerability: DNNs are susceptible to adversarial attacks [35], where minor, specifically designed perturbations to input images can significantly degrade their performance [4]. While early adversarial attack methods primarily relied on global perturbations [35, 15, 4] constrained by $\ell_2$ or $\ell_\infty$ norms and sparse perturbations [7, 24, 8] constrained by $\ell_0$ norm, recent approaches have

---

*corresponding authors: zuoxq@bupt.edu.cn

39th Conference on Neural Information Processing Systems (NeurIPS 2025).

shifted focus towards localized perturbations, known as adversarial patches [12, 42, 16]. While being more visually conspicuous, they have greater practical applicability and effectiveness [2, 40, 25].

Adversarial patches, characterized by distinct shapes, locations, number, and content, significantly complicate the attack optimization process due to their high-dimensional and combinational nature. Previous studies typically simplify this complex problem by independently optimizing patch content [20] or location [28, 41], often assuming fixed shapes and limiting patches to single or basic geometric forms [45, 27]. This simplification, however, considerably restricts the adversarial potential and practicality of these attacks, leaving open a critical research gap: **the joint optimization of multiple essential patch factors, including shapes, locations, number, and content.**

Motivated by this critical challenge, we propose **IMPACT** (Irregular Multi-Patch Adversarial Composition Based on Two-Phase Optimization), a flexible and novel adversarial attack framework specifically designed to support comprehensive optimization of multiple adversarial patch factors. The IMPACT framework is inherently versatile, allowing the integration of any gradient-free optimization algorithms tailored to specific adversarial objectives or constraints. To demonstrate the effectiveness and practicality of IMPACT, we present one concrete implementation leveraging evolutionary algorithms (EAs), renowned for their efficacy in black-box optimization tasks lacking gradient information. Our EA-based implementation of IMPACT introduces a two-phase optimization scheme:

**Joint Optimization Phase**: This phase jointly optimizes patch masks (defining shape and location) and patch content. To overcome the curse of dimensionality, we introduce a novel dimensionality reduction encoding scheme, reducing computational complexity while maintaining solution expressiveness. Additionally, we develop a random aggregation algorithm that plays a crucial role in generating practical adversarial patches. Unlike methods relying on fixed shapes [43, 36, 27], our algorithm produces diverse, irregular patch geometries by merging local regions. This ensures each patch is locally well formed, making them suitable for physical application. The irregularity further enhances their adversarial potential. This phase is implemented with differential evolution (DE).

**Refinement Phase**: This phase precisely refines patch content at the pixel level, transitioning from global exploration in phase 1 to targeted local exploitation, thus further optimizing attack effectiveness. In our implementation, this phase uses (1+1)-ES to balance computational cost and attack efficacy.

| Phase One | Phase Two | White-Box |
| :---: | :---: | :---: |

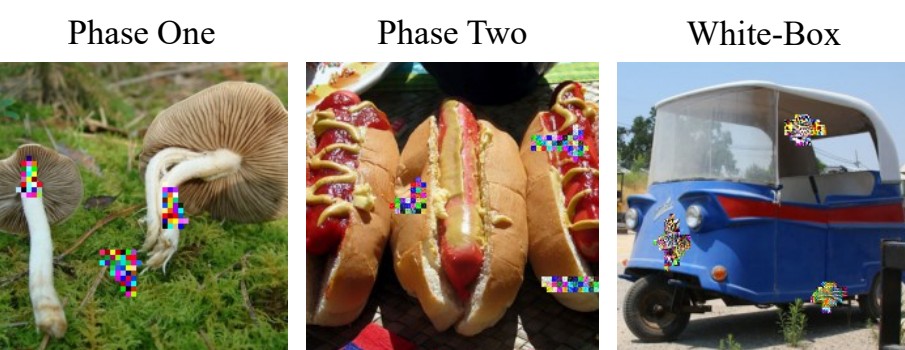

Figure 1: Adversarial patches by IMPACT. Phase One: DE-optimized block content. Phase Two: (1+1)-ES refined pixel content. White-Box: IMPACT mask with gradient-optimized content. Examples show diverse shapes, locations, and content.

Extensive experiments conducted on widely-used benchmark models (ResNet50 [17], VGG16 [32], ViT-B [11]) confirm that IMPACT outperforms multiple state-of-the-art patch attack methods in black-box scenarios. Our results underline the importance of jointly considering all key patch factors, demonstrating that IMPACT can significantly improve adversarial effectiveness. Figure 1 presents examples of adversarial patches generated by IMPACT. These patches exhibit locally coherent structures, making them physically printable and suitable for real-world deployment. Their irregular, optimized designs further enhance adversarial effectiveness by increasing the perturbation diversity and expressiveness. Our main contributions are summarized as follows:

- We introduce IMPACT, a flexible framework for adversarial patch attacks that allows seamless integration of diverse gradient-free optimization algorithms. The presented EA-

based implementation showcases the framework's capability for joint optimization of all critical adversarial patch factors.

- In line with the IMPACT framework, we propose a novel dimensionality reduction encoding scheme, simplifying the high-dimensional solution space while preserving the solution's diversity and quality. We further develop a new aggregation algorithm that merges scattered patch elements into locally coherent and irregular shapes, enabling physically printable adversarial patches.

- Through comprehensive evaluations against state-of-the-art methods, we demonstrate the superior effectiveness and efficiency of our IMPACT implementation. This systematic analysis provides valuable insights into how jointly optimizing multiple patch factors can substantially enhance adversarial attack success rates.

## 2    Related Work

This section provides an overview of two related research directions: Adversarial patch attacks, which craft localized, visible perturbations, and EA-based methods for black-box adversarial attacks.

### 2.1    Adversarial Patch Attacks

Depending on the level of knowledge an attacker has about the target model, adversarial patches can be categorized into *white-box attacks* and *black-box attacks*.

For the white-box attack, most studies focus primarily on optimizing patch content through gradient-based techniques. Brown et al. [2] proposed a universal adversarial patch for real-world targeted attacks. Karmon et al. [20] introduced LaVAN for generating visible localized noise patches. Their work focused on adversarial attacks in the digital domain, where modifications are made directly to the pixel values of digital images in a dataset. This approach enabled successful attacks using significantly smaller visible perturbations. Unlike earlier studies on universal patches designed to work across various images, Rao et al. [28] showed that image-specific patches, optimized for individual images, offer a more powerful alternative by leveraging the unique characteristics of each image. They emphasized the importance of patch location, proposing an optimization algorithm to determine the most effective location for the patch. Chen et al. [5] recognized that the patch shape is an equally critical factor. Building on this insight, they proposed the deformable patch attack. However, their approach was limited to generating a single patch. In the context of multi-patch attacks, Fu et al. [14] introduced PATCH-FOOL, which leverages an attention-aware patch selection mechanism to generate multiple patches simultaneously. Sharma et al. [31] further explored multi-patch attacks, demonstrating the advantages of using multiple patches over single-patch attacks. In addition, Huang et al. [19] proposed a multi-mini-patch adversarial attack for remote sensing image classification. However, these multi-patch schemes are restricted to rectangular shaped patches.

For the black-box attack, random search is the dominating approach for optimizing patch content. Fawzi et al. [13] were among the first to explore patch-based black-box attacks. Their method can generate rectangular, monochromatic patches with optimized shape and placement. Unfortunately, the attack's effectiveness was limited due to the simplicity of the patch content. To address this issue, Yang et al. [45] proposed TPA, where reinforcement learning was employed to optimize both the position and texture parameters of each patch. Croce et al. [8] introduced a robust adversarial attack framework Patch-RS based on random search. However, the patches were limited to fixed square shapes. To investigate the impact of different shapes, Ran et al. [27] proposed a cross-shaped adversarial patch, consisting of two intersecting line segments extending toward the corners of the input image. Using random search to optimize position and content, the method achieved high attack success rate. However, the global perturbation structure caused the line segments to become excessively thin, rendering it challenging to apply in the physical world.

In summary, existing adversarial patch attack methods predominantly focused on optimizing one or two factors, such as patch shape, location, or content, while *neglecting a comprehensive, joint optimization of all critical factors*. This narrow focus restricts their capacity to fully leverage the optimization space, highlighting the potential for more comprehensive and effective approaches.

## 2.2 Adversarial Attacks Based on EAs

In the field of adversarial attacks, EAs have demonstrated its unique strength of effectively optimizing arbitrary target models without using gradient information [10, 43, 36, 22]. Several studies [1, 26, 22] have investigated the application of EA to generate adversarial examples under $\ell_2$ and $\ell_\infty$ constraints. Other works [33, 38, 37] have focused on sparse adversarial attacks under $\ell_0$ constraints.

Very recently, EAs have been increasingly used to generate adversarial patches; however, existing encoding schemes in these approaches do not support full optimization across all patch factors. For example, Williams et al. [43] introduced CamoPatch as an EA approach to generate camouflaged adversarial patches. Unlike traditional adversarial patches, this approach focuses on reducing the patch's visibility. Hu et al. [18] proposed AdvIB, leveraging DE to create adversarial patches deployable in the physical world. However, due to the limitations of their encoding scheme, their method only supports rectangular patch shapes and monochromatic patch content. Tang et al. [36] introduced a dimensionality reduction strategy focused on patch content, leveraging duplicating and tiling to upscale decision variables from a low-dimensional space to a higher-dimensional space. While this approach enhances the optimization process for patch content, their encoding scheme has notable limitations. Specifically, it only addresses patch content optimization while ignoring the patch mask, which remains in fixed square shapes.

Overall, most existing methods remain focused on designing encoding schemes to optimize patch content, while a *comprehensive approach that integrates patch shape, location, number, and content into the encoding process is still lacking*. This gap motivated us to develop a novel encoding scheme that supports joint optimization across all these critical patch factors.

# 3 Proposed Method

We present IMPACT, a framework that simultaneously optimizes both the patch mask and its content. Specifically, the IMPACT framework comprises two optimization phases. In phase 1, IMPACT operates at a block-level to simultaneously optimize the patch mask and the patch content. This phase aims for broad exploration of the solution space to identify promising patch configurations. Following phase 1, phase 2 focuses on meticulous, pixel-level optimization of the patch content. In this section, we first formulate the problem of adversarial patch attacks. Subsequently, we present the detailed EA-based implementation of IMPACT. Algorithm 1 outlines the overall procedure of using EA-based IMPACT for black-box adversarial patch attacks. Descriptions of the key functions within Algorithm 1 are provided in Appendix A.

---

**Algorithm 1** Irregular Multi-Patch Adversarial Attack Based on Two-Phase Optimization

---

**Input:** Model $f$, original example $x$, true label $y$, number of mini-patches $n$, number of patches $k$, population size $N$, DE iterations $T_d$, (1+1)-ES iterations $T_e$

**Output:** Adversarial example $\hat{x}$

    **// Phase 1: DE for Joint Optimization**
1: Initialize population $P_0 \leftarrow \text{PopInit}(n, k, N)$
2: Compute initial fitness $F_0 \leftarrow \text{Fitness}(P_0, f, x, y)$
3: **for** $t = 1, \ldots, T_d$ **do**
4:     $V_t \leftarrow \text{Mutation}(P_{t-1})$
5:     $U_t \leftarrow \text{Crossover}(P_{t-1}, V_t)$
6:     $U_t \leftarrow \text{Aggregation}(U_t, k)$
7:     $F_u \leftarrow \text{Fitness}(U_t, f, x, y)$
8:     $P_t, F_t \leftarrow \text{Selection}(P_{t-1}, F_{t-1}, U_t, F_u)$
9:     $p^* \leftarrow \text{SelectBest}(P_t, F_t)$
10:    Construct patches $(\delta, M) = \text{BuildPatch}(p^*)$
11:    Generate $\hat{x}$ using $(\delta, M)$ according to Eq. (1)
12:    **if** $f(\hat{x}) \neq y$ **then**
13:       successful = True

14:       **break**
15:    **end if**
16: **end for**

    **// Phase 2: (1+1)-ES for Content Refinement**
17: **for** $t = 1, \ldots, T_e$ **do**
18:    **if** successful = True **then**
19:       **break**
20:    **end if**
21:    Add Gaussian noise $\delta_{\text{noise}}$: $\delta' = \delta + \delta_{\text{noise}}$
22:    Generate $\hat{x}'$ using $(\delta', M)$
23:    **if** $\text{fitness}(\hat{x}') > \text{fitness}(\hat{x})$ **then**
24:       Update $\delta = \delta', \hat{x} = \hat{x}'$
25:    **end if**
26:    **if** $f(\hat{x}) \neq y$ **then**
27:       successful = True
28:    **end if**
29: **end for**
30: **return** Adversarial example $\hat{x}$

---

### 3.1 Problem Formulation

Given an original example $x \in \mathcal{R}^{c \times h \times w}$, the objective of adversarial patch attacks is to create an adversarial example $\hat{x} \in \mathcal{R}^{c \times h \times w}$ that can mislead the model into making incorrect predictions [35]. Here, $c$, $h$, and $w$ correspond to the number of channels, height, and width of the example, respectively. An adversarial patch consists of two components: a mask $M \in \{0,1\}^{c \times h \times w}$, which determines the shape and location of the patch, and a perturbation $\delta \in \mathcal{R}^{c \times h \times w}$, which defines the patch content. By combining $x$, $M$, and $\delta$, the adversarial example $\hat{x}$ can be defined as follows:

$$\hat{x} = x \odot (1 - M) + \delta \odot M, \tag{1}$$

where $\odot$ is the element-wise Hadamard product. As a result, the perturbation is applied in regions with $M_{ij} = 1$, while regions with $M_{ij} = 0$ retain the original image content.

Performing adversarial patch attacks requires solving the following optimization problem [27]:

$$\arg\min_{\delta, M} \mathcal{L}(f(x \odot (1 - M) + \delta \odot M), \hat{y}), \text{s.t.} \|M\|_0 < \epsilon, \tag{2}$$

where $f$ denotes the image classification model, $\mathcal{L}$ is its loss function, and $\|M\|_0 < \epsilon$ imposes an $\ell_0$-norm constraint to limit the patch area. For the untargeted attack, $\hat{y}$ can be any label other than the original label $y$. For the targeted attack, $\hat{y}$ is set to the target label $y_t$.

The problem defined in Equation (2) is a joint optimization problem where $M$ and $\delta$ jointly define the optimization solution space. Many existing adversarial patch attack methods simplify the problem by decoupling these two components [20, 42]. Methods that pursue a truly joint optimization of both mask and perturbation remain conspicuously absent.

### 3.2 Phase 1: Joint Optimization

Phase 1 of IMPACT tackles the challenging problem of jointly optimizing the patch mask and content, which defines a complex and high-dimensional search space [36]. Direct pixel-level optimization is computationally prohibitive due to the sheer number of variables involved. Additionally, the binary nature of the mask often leads to fragmented, incoherent shapes that are not physically realizable [42]. To overcome these challenges, we introduce two key innovations: a dimensionality reduction encoding scheme to compress the solution space, and a new aggregation algorithm to ensure locally coherent and contiguous patch shapes. Within this framework, we employ DE as the core optimization engine, leveraging its proven effectiveness in black-box adversarial settings for efficiently exploring complex solution spaces [36, 6].

#### 3.2.1 Dimensionality Reduction Encoding

We develop a novel dimensionality reduction method that significantly reduces the encoding length while addressing limitations in existing approaches. Unlike traditional methods [43, 36], our encoding method supports the first time joint optimization of all critical patch factors. Such individual representation not only enhances optimization efficiency but also enables a more comprehensive and effective exploration of the adversarial patch design space.

In our framework, the first phase of optimization utilizes DE, which is a population-based algorithm. This means it maintains a population, which is a set of candidate solutions. We use $p_i$ to represent the $i$-th individual in the population, where $i \in [1, N]$ and $N$ denotes the population size. Each individual $p_i$ in the population represents a candidate solution that encodes all the necessary information to generate the patches. Concretely, each individual $p_i$ is encoded into two parts: $p_i = (\mathbf{m}_i, \mathbf{r}_i)$. The first component $\mathbf{m}_i$ represents the mask $M$, and the second part $\mathbf{r}_i$ represents the perturbation $\delta$. The binary array $\mathbf{m} = [b_1, b_2, ..., b_l]$ is used to encode the mask $M$, where $b_i \in \{0, 1\}$ and $l$ is the encoding length of

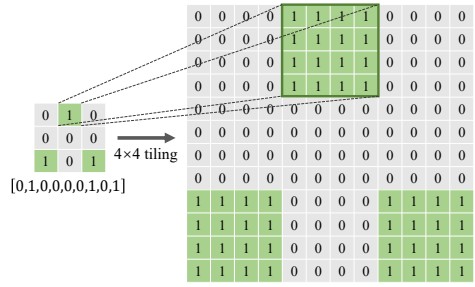

Figure 2: Example of mask encoding. A 9-element binary array $[0, 1, 0, 0, 0, 0, 1, 0, 1]$ can represent a $12 \times 12$ mask.

$\mathbf{m}$, depending on the size $h \times w$ of the original mask. When we set $l = (h/4) \times (w/4)$, each element in $\mathbf{m}$ corresponds to a $4 \times 4$ block in the mask $M$. We can use a method called "$4 \times 4$ tiling" to reconstruct $\mathbf{m}$ back to the original size of the mask. An example of this encoding method is shown in Figure 2. This method significantly compresses the optimization search space and reduces its complexity. For the $224 \times 224$ images in the ImageNet dataset, the mask $M$ can be reduced to a size of $56 \times 56$. For the encoding of the perturbation $\delta$, we use a three-channel matrix $\mathbf{r} \in \mathcal{R}^{3 \times n}$, where $n$ is the number of one-valued elements in $\mathbf{m}$ and $\mathcal{R}$ is constrained to $[0, 255]$. This design exploits the fact that the perturbation is applied only at positions where $M_{ij} = 1$ according to Equation (1). Hence optimization is restricted to those specific pixel locations. Notably, each element in $\mathbf{m}$ represents a $4 \times 4$ mini-patch, and each three-channel pixel in $\mathbf{r}$ encodes the color information of the corresponding $4 \times 4$ patch.

### 3.2.2 Random Aggregation

After defining our encoding scheme, the resulting encoded mask $\mathbf{m} \in \{0, 1\}^l$ will contain $n$ active elements. However, these elements may be spatially dispersed. To address this issue, we propose a new aggregation algorithm to transform $\mathbf{m}$ into a new mask $\hat{\mathbf{m}}$ where these $n$ active elements are consolidated into $k$ locally connected, and irregular patches. The algorithm proceeds as follows:

The 1D input mask encoding $\mathbf{m} \in \{0, 1\}^l$ is first reshaped into a 2D binary matrix $M' \in \{0, 1\}^{\sqrt{l} \times \sqrt{l}}$ for spatial processing. The set of coordinates of the $n$ active elements in $M'$ is extracted:

$$\mathcal{X} = \{\mathbf{p}_j = (x_j, y_j) \mid M'[\mathbf{p}_j] = 1, j = 1, \ldots, n\}. \tag{3}$$

These coordinates $\mathcal{X}$ are then partitioned into $k$ clusters using the K-Means algorithm:

$$\mathcal{C} = \{C_1, C_2, \ldots, C_k\} = \text{KMeans}(\mathcal{X}, k). \tag{4}$$

The aggregation process aims to form a single connected shape using the elements assigned to $C_i$. For each cluster, we randomly select a point within the cluster as the aggregation center. For each point in a cluster, we select a target point uniformly at random from the aggregation center's 8-neighborhood and move the point toward that target point, repeating this process until the point becomes adjacent to the existing connected component. During each move, we randomly choose whether to prioritize a horizontal or vertical move. The randomness in selecting aggregation center, the target point, and the order of move attempts contributes to the diversity of the resulting aggregated shapes. A detailed description of the random aggregation algorithm (Algorithm 2) along with visual explanation can be found in Appendix B.

Powered by the newly developed encoding scheme and the random aggregation algorithm, we further adopt DE to optimize the patch mask and content in phase 1. The detailed design of our DE algorithm is provided in Appendix C.

### 3.3 Phase 2: Content Refinement

Phase 2 aims to further improve attack effectiveness through pixel-level refinement of the patch content, using the mask and initial content established in Phase 1. This refinement is particularly important when phase 1 does not yield a successful attack. The solution candidate in this phase is represented by the full-resolution perturbation $\delta$, which has dimensions $3 \times h \times w$, matching the input image resolution. We employ the (1+1) -ES for fine-grained content refinement in this phase, because it is notably simple to implement, and computationally efficient due to evaluating only one candidate solution per iteration. **Note that the IMPACT framework is flexible, and any black-box optimizers could be employed for both phases**.

The initial $\delta$ are inherited from phase 1. In phase 2, $M$ remains fixed while $\delta$ is refined through iterative application of Gaussian noises. At each iteration, noise $\delta_{\text{noise}}$ sampled from $\mathcal{N}(0, \sigma^2)$, where $\sigma$ controls the perturbation magnitude, is added to the current $\delta$. The updated perturbation is then applied to the input image using the fixed mask $M$ to generate a new adversarial example. If the new perturbation improves fitness, it replaces the current best solution. This process continues until a successful attack is achieved or the maximum number of iterations is reached. By transitioning from block-wise to pixel-level optimization, phase 2 overcomes the limitations of using coarse $4 \times 4$ perturbations, enabling precise adjustments that significantly enhance the attack's effectiveness.

# 4 Experiments

In this section, we conduct experimental evaluation to assess the effectiveness of our proposed IMPACT method. Section 4.1 provides a detailed overview of the experimental setup. Section 4.2 presents a comparative analysis of IMPACT against state-of-the-art patch-based attack methods. Section 4.3 offers ablation studies to analyze the contribution of key components. Finally, Section 4.4 explores the significance and interplay of various hyperparameters in our method.

## 4.1 Experimental Setup

**Dataset and Models**: Following previous works [8, 43, 27], we use ImageNet [9] for evaluation due to its diverse object categories and real-world scenarios, enabling a comprehensive assessment of our method's effectiveness. Additional experiments on more datasets are provided in Appendix E.5. Following the same setup as Patch-RS [8], we randomly select a subset of 500 images from the validation set of ImageNet for our experiments. For the victim models, we employ three widely adopted architectures: ResNet50 [17], VGG16 [32], and ViT-B [11]. These models encompass diverse architectural designs, enabling a comprehensive evaluation of IMPACT's effectiveness across varying network architectures. All input images are resized to a standard size of $224 \times 224$, consistent with the requirements of the experimented models. The models are officially pre-trained on the full ImageNet training set, ensuring a robust and reliable baseline for evaluation. All experiments were conducted on a system equipped with an NVIDIA GeForce RTX 4090 GPU. The detail parameter settings are provided in Appendix E.1.

**Evaluation Metrics**: We use the attack success rate (ASR) as the evaluation metric, considering only input images the deep model classifies correctly in the absence of any attack. For untargeted attacks, ASR measures the proportion of such images where the adversarial patch successfully causes the model to misclassify them. For targeted attacks, ASR evaluates the percentage of input images where the adversarial patch forces the model to classify the input into a specific, pre-defined target class. For query efficiency, we utilize the average query count (AQ) to measure the average number of queries the attack algorithm requires to successfully craft adversarial patches. Definitions of these performance metrics are presented in Appendix E.2. In addition, due to the stochastic nature of IMPACT, we use the same group of random seeds for evaluations to ensure reproducibility. The influence of multiple seeds is further discussed in Appendix E.3.

## 4.2 Performance Comparison

To evaluate our proposed IMPACT method, we conduct comparisons with state-of-the-art adversarial patch attack methods, including Patch-RS [8], TPA [45], Patch-Fool [14]. Our primary evaluation focuses on IMPACT's performance in challenging black-box scenarios. Additionally, Appendix E.4 presents a comparison between the white-box variant of IMPACT and Patch-Fool. Furthermore, to assess its robustness, IMPACT's effectiveness against various common defense mechanisms is evaluated in Appendix E.6. Appendix E.7 provides the results of physical-world experiments, further demonstrating the practicality of IMPACT. Appendix E.8 reports a detailed runtime analysis. Below, we present a detailed analysis of our experimental comparison results.

For the black-box comparison, Table 1 presents the statistical results of adversarial attacks conducted on various ImageNet classification models. Here, The Query represents the query budgets, and 1%, 2% are the percentages of perturbation areas. The ASR is expressed as a percentage, and for simplicity, we have omitted the unit in the Table 1. We select Patch-RS and TPA as baselines, as they focus on optimizing patch content and location, although their shapes remain fixed as rectangles.

The experimental results consistently demonstrate the superiority of our IMPACT method, which achieves higher ASR and generally lower AQ compared to the baselines across different models and settings. As shown in Table 1, IMPACT exhibits strong performance in untargeted scenarios. For instance, when attacking ResNet50 with a query budget of 10,000 and a 2% perturbation area, our method achieves a maximum ASR of 96.4%, while Patch-RS achieves 93.6%. Additionally, our AQ is 1044, which is better than Patch-RS's 1408. For targeted attacks, our method demonstrates even greater advantages. On the ResNet50 model, we achieve a maximum targeted ASR of 57.8%, whereas Patch-RS only reaches 20.0%. Moreover, our AQ is 7148, which is significantly smaller than Patch-RS's 8885. We attribute IMPACT's enhanced effectiveness primarily to its joint optimization

Table 1: Performance comparison for black-box adversarial patch attacks.

| Model | Query | Method | Untargeted Attack | | | | Targeted Attack | | | |
|---|---|---|---|---|---|---|---|---|---|---|
| | | | 1% | | 2% | | 1% | | 2% | |
| | | | ASR | AQ | ASR | AQ | ASR | AQ | ASR | AQ |
| **ResNet50** | 5000 | IMPACT | **87.2** | **1236** | **94.2** | **676** | **24.6** | **4379** | **38.4** | **4126** |
| | | Patch-RS | 82.4 | 1360 | 89.8 | 982 | 7.6 | 4790 | 12.2 | 4711 |
| | | TPA | 38.0 | 3519 | 51.0 | 2807 | 4.3 | 4888 | 7.8 | 4781 |
| | 10000 | IMPACT | **90.0** | **1518** | **96.4** | **1044** | **38.4** | **8239** | **57.8** | **7148** |
| | | Patch-RS | 88.2 | 1990 | 93.6 | 1408 | 10.8 | 9359 | 20.0 | 8885 |
| | | TPA | 51.0 | 5705 | 57.0 | 5091 | 7.6 | 9730 | 15.4 | 9479 |
| **VGG16** | 5000 | IMPACT | **92.8** | **862** | **94.2** | 841 | **16.2** | **4635** | **34.6** | **4327** |
| | | Patch-RS | 88.6 | 1114 | 92.8 | **832** | 10.8 | 4704 | 15.6 | 4640 |
| | | TPA | 42.2 | 3416 | 53.6 | 2686 | 5.6 | 4880 | 8.2 | 4805 |
| | 10000 | IMPACT | **94.6** | **1310** | **95.2** | **1069** | **27.6** | **8893** | **35.4** | **8295** |
| | | Patch-RS | 92.4 | 1562 | 94.4 | 1191 | 16.6 | 9073 | 30.8 | 8425 |
| | | TPA | 56.2 | 5204 | 59.8 | 4777 | 9.4 | 9598 | 16.8 | 9321 |
| **ViT-B** | 5000 | IMPACT | **85.6** | **1315** | **92.8** | **880** | **18.4** | **4552** | **30.2** | **4311** |
| | | Patch-RS | 80.2 | 1457 | 87.4 | 1052 | 6.2 | 4825 | 10.6 | 4786 |
| | | TPA | 35.6 | 3607 | 48.2 | 2956 | 3.8 | 4942 | 6.2 | 4855 |
| | 10000 | IMPACT | **88.4** | **1652** | **95.0** | **1157** | **30.6** | **8553** | **48.2** | **7552** |
| | | Patch-RS | 86.2 | 2175 | 91.8 | 1524 | 9.2 | 9456 | 17.8 | 9057 |
| | | TPA | 48.4 | 5854 | 55.2 | 5237 | 6.8 | 9828 | 13.6 | 9553 |

of multiple patch factors, including irregular shapes, locations, number, and content. This holistic approach allows IMPACT to explore a more expressive solution space, enabling the discovery of more potent adversarial patches. For a deeper understanding of how IMPACT achieves this by influencing the model's internal mechanisms, we provide an effectiveness analysis in Appendix D.

## 4.3 Ablation Study

To assess IMPACT's key components, we perform ablations on DE, (1+1)-ES, and dimensionality reduction encoding. Note that we do not ablate the random aggregation algorithm because it is essential for transforming sparse modifications into patch-shaped perturbations, and removing it would change the attack type. However, to validate its design, we investigate the impact of the stochastic elements within this algorithm itself in Appendix E.9. All ablation experiments were performed on the ImageNet against the ResNet50 model under a total query budget of 5000, with a 2% patch area distributed across 3 patches. Similar trends were observed on other model architectures.

Table 2: Ablation study on components of IMPACT.

| Method | Phase 1 | Phase 2 | ASR | AQ |
|---|---|---|---|---|
| IMPACT | DE | (1+1)-ES | **94.2** | **676** |
| *Effectiveness of DE in Phase 1:* | | | | |
| (1+1)-ES-only | None | (1+1)-ES | 60.9 | 2269.82 |
| RS + (1+1)-ES | RS | (1+1)-ES | 78.4 | 1698.7 |
| GA + (1+1)-ES | GA | (1+1)-ES | 81.8 | 1523.8 |
| *Effectiveness of (1+1)-ES in Phase 2:* | | | | |
| DE-Only | DE | None | 90.2 | 748 |
| DE + RS | DE | RS | 92.3 | 842.1 |

Table 3: Encoding granularity ablation. For each tile size $t \times t$, the number of mini-patches was adjusted to maintain the total patch area fixed at 2% of the image.

| Tile Size | Mini-Patch | ASR | AQ |
|---|---|---|---|
| $1 \times 1$ | 1024 | 87.1 | 751.5 |
| $2 \times 2$ | 256 | 89.2 | 691 |
| $4 \times 4$ | 64 | **94.2** | **676** |
| $8 \times 8$ | 16 | 86.3 | 822.5 |
| $16 \times 16$ | 4 | 80.5 | 1099.5 |
| $32 \times 32$ | 1 | 57.2 | 1824 |

Table 2 presents the results comparing our full IMPACT framework against several variants designed to assess the roles of its constituent optimizers and the two-phase structure. To validate the choice of DE, we compared IMPACT against variants where DE was replaced by Random Search (RS) and Genetic Algorithm (GA). These results underscore the superior exploratory capabilities of DE in navigating the complex, high-dimensional search space of joint mask and content optimization. To demonstrate the necessity of the second refinement phase, we compare IMPACT with a DE-Only variant and DE+RS. In DE-Only, the entire 5000 query budget is allocated to the DE algorithm for

joint mask and content optimization, omitting the (1+1)-ES refinement. As shown, IMPACT achieves an ASR of 94.2% with 676 AQ, whereas DE-Only only reaches 90.2% ASR with 748 AQ. This improvement highlights that the fine-grained adjustments performed by (1+1)-ES in Phase 2 are crucial for converting near-successful patches into effective adversarial examples.

Furthermore, IMPACT employs a $4 \times 4$ tiling strategy as the default for its dimensionality reduction encoding scheme. To validate this choice and understand the impact of different encoding granularities, we conducted an encoding granularity ablation by varying the tile size. The results are presented in Table 3. Using smaller tiles resulted in a higher-dimensional search space. While offering finer granularity, this increased complexity led to slightly lower ASR. Moreover, excessively small tile sizes incur prohibitively long optimization times. Using larger tiles can significantly reduce the dimensionality. However, this led to a marked decrease in ASR. This indicates that the coarse granularity severely limited the ability to form effective adversarial patches. These findings confirm that the 4×4 tiling offers an effective balance between dimensionality and spatial resolution to construct diverse and potent irregular adversarial patches.

## 4.4 Parameter Sensitivity Analysis

Our method involves five important parameters that can be adjusted: $n$, $k$, $N$, $T_d$, and $T_e$. Here $n$ is the number of $4 \times 4$ mini-patches in the mask, controlling the perturbation area. $k$ denotes the number of patches. $N$ represents the population size. $T_d$ and $T_e$ refer to the iterations of the DE and (1+1)-ES algorithm, respectively. Different parameter settings can lead to varying attack effectiveness.

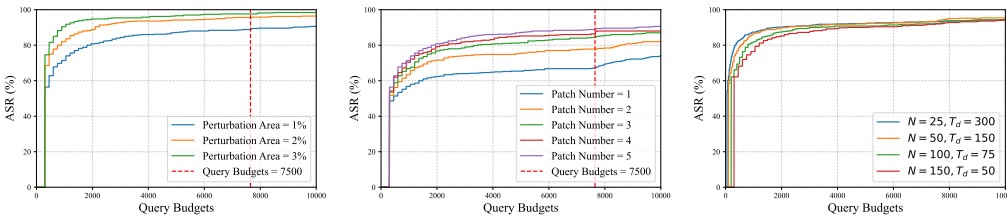

Figure 3: ASR vs query budgets. Effect of perturbation areas (Left). Effect of patch number (Center). Effect of $N$ and $T_d$ (Right).

To better illustrate the impact of different parameters, we plot the variation curves of success rates under different query budgets in Figure 3. The results demonstrate that a larger perturbation area and a greater number of patches contribute to higher ASR. Moreover, while different combinations of $N$ and $T_e$ show similar performance under high query budgets, smaller populations with more iterations excel under lower budgets. This suggests that prioritizing iteration count over population size can improve efficiency, especially with limited query resources. In addition to these primary analyses, we conducted further parameter experiments, including an investigation into the impact of DE's mutation factor and crossover probability. These results demonstrate the relative robustness of IMPACT to these parameters. Detailed experimental data for those studies can be found in Appendix E.10.

Our IMPACT supports generating multiple patches simultaneously. To demonstrate the improvement in attack effectiveness achieved by using multiple patches, we conduct an experiment to assess the impact of varying the number of patches. As depicted in Figure 4, across all tested total perturbation areas, increasing the number of patches generally leads to a noticeable improvement in ASR. This trend highlights a key advantage of multi-patch strategies: by distributing adversarial perturbations across multiple smaller, strategically placed regions, IMPACT is able to degrade model accuracy more effectively than using a single concentrated patch. This ability to alter diverse local regions within the image significantly enhances the overall attack effectiveness.

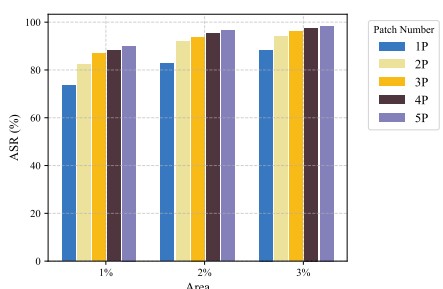

Figure 4: Impact of the number of patches ($k$) on ASR for different total perturbation areas ($1\%, 2\%, 3\%$). Experiments were conducted on ResNet50 with $N = 150, T_d = 50, T_e = 2500$.

## 5 Conclusion

In this paper, we introduced a novel irregular multi-patch adversarial attack framework that supports simultaneously optimizing all critical patch factors, including shape, location, number, and content. Under our IMPACT framework, DE was used first to jointly optimize the patch mask and content, supported by a unique dimensionality reduction encoding scheme. Secondly, (1+1)-ES further refined the patch content for improved precision. Additionally, we proposed a random aggregation algorithm to generate diverse, irregular patch designs for practical use. Extensive experiments demonstrated that our method outperformed state-of-the-art approaches, significantly enhancing attack success rates. This work provides valuable insights into adversarial patch design and optimization, paving the way for practical methodologies in this domain.

**Limitations:** This work focuses on score-based black-box attacks, where access to the model's output probabilities enables effective guidance of the evolutionary optimization. In contrast, decision-based black-box attacks, which provide only the model's top-1 predicted label, pose a fundamentally different and more restrictive challenge. The absence of probability scores makes it significantly harder to infer meaningful search directions, especially when jointly optimizing multiple patch factors. Addressing this setting would require substantially different algorithmic designs and is therefore beyond the scope of this paper. Nevertheless, extending IMPACT to support decision-based attacks remains an important direction for future research. Moreover, as IMPACT is primarily an empirical approach, a more rigorous theoretical analysis of its underlying mechanisms constitutes an important direction for future research.

**Ethics Statement:** This work introduces IMPACT, a framework for crafting irregular multi-patch adversarial attacks to uncover vulnerabilities in deep vision models. While these attacks are essential for developing stronger defenses and improving model robustness, they also carry the risk of malicious misuse. We advocate for the responsible and ethical application of such technologies, emphasizing their use as tools for the advancement of trustworthy AI, rather than for purposes that could cause harm or compromise system integrity.

## Acknowledgments

This work was supported in part by the National Natural Science Foundation of China (No.62272117 and No.62476030).

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

# Appendix

## A  Function Explanation

To provide additional clarity, we summarizes the functions used in Algorithm 1. Each function plays a specific role in the EA-based optimization framework of IMPACT.

**PopInit:** This function initializes a population $P_0$ of $N$ individuals to ensure diverse starting points for the search process. Each individual is structured according to our encoding scheme, consisting of two components: a binary encoded mask $\mathbf{m}$ and a continuous encoded content $\mathbf{r}$.

**Mutation:** This core DE function creates a mutant vector $v_i$ for each individual $p_i$ by differential combination of other population members. This process introduces new variations into the search.

**Crossover:** This function mixes the components of the parent individual $p_i$ and its corresponding mutant vector $v_i$ to create a new trial individual $u_i$. This enhances population diversity.

**Aggregation:** This is a component of our framework. It takes the sparse binary mask $m$ from a trial individual and transforms its scattered active elements into $k$ physically contiguous, irregular patches.

**Fitness:** This function evaluates the quality of a trial individual. It reconstructs the full patch, applies it to the image, and calculates the model's cross-entropy loss.

**Selection:** This function compares the fitness of the trial individual $u_i$ with the parent $p_i$. The one with the better fitness score survives into the next generation's population.

**SelectBest:** This is a simple function that iterates through the final population and returns the single individual with the best fitness score found during the optimization.

## B  Random Aggregation Details

To provide a more detailed exposition of our random aggregation algorithm, we present it in Algorithm 2. This algorithm is a crucial component of our IMPACT framework, responsible for transforming the spatially dispersed active mini-patches into coherent, irregular patch structures.

---

**Algorithm 2** Random Aggregation Algorithm

---

**Input:** Mask encoding $\mathbf{m} \in \{0,1\}^l$, patch number $k$
**Output:** Aggregated mask encoding $\hat{\mathbf{m}}$

1: Reshape $\mathbf{m}$ into a 2D binary matrix $\hat{M} \in \{0,1\}^{\sqrt{l} \times \sqrt{l}}$.
2: Extract coordinates of one-valued elements: $\mathcal{X} = \{(a,b) \mid \hat{M}[a,b] = 1\}$.
3: Perform clustering: $\mathcal{C} = KMeans(\mathcal{X}, k)$, where $\mathcal{C} = \{C_1, C_2, \ldots, C_k\}$.
4: **for** each $C_i \in \mathcal{C}$ **do**
5:     Select $s_{\text{center}} \in C_i$ uniformly at random.
6:     Create aggregated region $A(s_{\text{center}})$.
7:     **for** each $s \in C_i \setminus \{s_{\text{center}}\}$ **do**
8:         Sample $s_{\text{target}} \in Nb(s_{\text{center}})$, where $Nb(s_{\text{center}})$ is the neighborhood of $s_{\text{center}}$.
9:         $s_{\text{new}} \leftarrow \text{Move}(s, s_{\text{target}})$.
10:        **if** $s_{\text{new}} \in Nb(A(s_{\text{center}}))$ **then**
11:            Update $A(s_{\text{center}}) \leftarrow A(s_{\text{center}}) \cup \{s_{\text{new}}\}$.
12:            Update $\hat{M}$: $\hat{M}[s] \leftarrow 0$, $\hat{M}[s_{\text{new}}] \leftarrow 1$.
13:        **end if**
14:     **end for**
15: **end for**
16: Flatten $\hat{M}$ to $\hat{\mathbf{m}} \in \{0,1\}^l$.
17: **return** $\hat{\mathbf{m}}$

---

Furthermore, to visually illustrate the algorithm's operation, Figure 5 depicts an example of the aggregation process. In this case, the initially scattered active elements are first partitioned into three distinct clusters by K-Means. Subsequently, our random aggregation algorithm processes each cluster, resulting in the formation of three distinct patches, each exhibiting an irregular and locally connected shape.

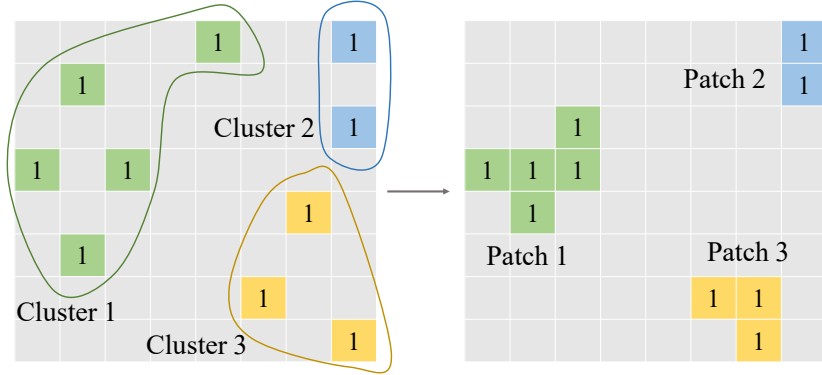

Figure 5: Example of random aggregation.

Figure 6 further demonstrates the practical outcome of this aggregation process. As can be seen, the algorithm effectively consolidates initially scattered units into locally connected, irregular shapes, thereby forming patches that are well-formed and readily applicable for adversarial attacks.

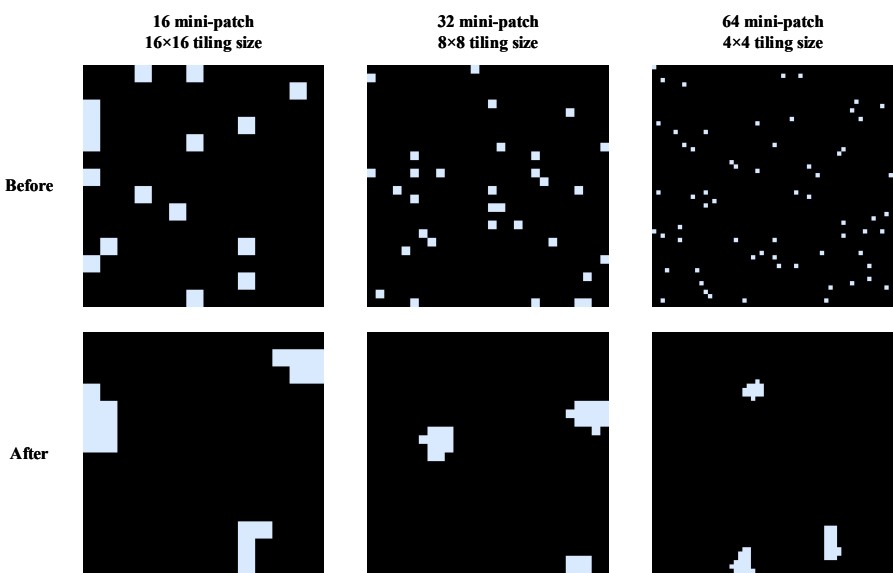

Figure 6: Visualization of the Random Aggregation algorithm's effect. The top row shows the spatial distribution of active mini-patches before aggregation, for different total mini-patch counts and corresponding tiling sizes. The bottom row illustrates how these scattered mini-patches are consolidated into coherent, irregular patch structures by our aggregation algorithm. Each column represents a different configuration. Left: 16 mini-patches and 16x16 tiling size. Center: 32 mini-patches and 8x8 tiling size. Right: 64 mini-patches and 4x4 tiling size.

## C  DE Components

### C.1  Initialization

The initial population $P_0$ for the DE algorithm, consisting of $N$ individuals, is generated randomly to ensure diverse starting points for the search process. Each individual $p \in P_0$ is structured according to our dimensionality reduction encoding scheme, consisting of two components: a binary encoded mask $\mathbf{m}$ and a continuous encoded content $\mathbf{r}$.

The mask $\mathbf{m}$ is a binary vector of length $l$. To satisfy the constraint on the total number of active mini-patches, exactly $n$ elements of $\mathbf{m}$ are randomly selected and set to 1, while the remaining $l - n$

elements are set to 0. This ensures that every individual in the initial population adheres to the specified patch area from the beginning of the optimization.

The content component $\mathbf{r}$ corresponds to the initial color information for the $n$ active mini-patches in its associated mask $\mathbf{m}$. It is represented as a matrix of shape $3 \times n$. The values within this matrix are initialized by sampling uniformly at random from the valid pixel intensity range $[0, 255]$.

This random initialization process generates a diverse population $P_0 = \{p_1, \ldots, p_N\}$ where each individual satisfies the patch area requirements and possesses varied initial content, providing a robust foundation for the subsequent evolutionary operations of mutation, crossover, and selection.

## C.2 Mutation

Building upon our dimensionality reduction encoding, we further tailor the core DE operators. In particular, mutation must be adapted to handle the distinct binary nature of the encoded mask and the continuous nature of the encoded content. We present these customized mutation procedures below.

For each individual $p_i$, we first randomly select three distinct individuals $p_a = (\mathbf{m}_a, \mathbf{r}_a)$, $p_b = (\mathbf{m}_b, \mathbf{r}_b)$, and $p_c = (\mathbf{m}_c, \mathbf{r}_c)$ from the population. These individuals serve as the basis for generating the mutated individual $v_i = (\tilde{\mathbf{m}}_i, \tilde{\mathbf{r}}_i)$, as detailed below. The mutation operation is applied separately to the two components of the individual.

For the mask component, we adopt a mutation strategy in the binary space. Using the selected mask components $\mathbf{m}_a$, $\mathbf{m}_b$, and $\mathbf{m}_c$ of the chosen individuals, the mutated mask $\tilde{\mathbf{m}}$ is computed as follows:

$$\tilde{\mathbf{m}}_i = (\mathbf{m}_a + (\mathbf{m}_b - \mathbf{m}_c)) \bmod 2. \tag{5}$$

Here, the modulo operation ensures that the resulting $\tilde{\mathbf{m}}_i$ remains a valid binary vector.

For the perturbation component in the continuous space, using the selected perturbation components $\mathbf{r}_a$, $\mathbf{r}_b$, and $\mathbf{r}_c$, the mutated component $\tilde{\mathbf{r}}_i$ is calculated as:

$$\tilde{\mathbf{r}}_i = \mathbf{r}_a + F \cdot (\mathbf{r}_b - \mathbf{r}_c), \tag{6}$$

where $F$ is a mutation factor. The crossover and selection mechanisms, which follow standard DE practices (e.g., binomial crossover and greedy selection), along with the detailed fitness function calculation, are described below.

## C.3 Crossover

Each mutated individual $v_i$ proceeds to the crossover operation, which is performed as follows:

$$u_{i,j} = \begin{cases} v_{i,j}, & \text{if } \mathrm{rand}_j(0,1) < CR \text{ or } j = j_{\mathrm{rand}} \\ p_{i,j}, & \text{otherwise} \end{cases}, \tag{7}$$

where $v_{i,j}$ is the $j$-th element of the mutant individual $v_i$. $p_{i,j}$ is the $j$-th element of the parent individual $p_i$. The crossover operation combines $v_i$ and $p_i$ to create a trial individual $u_i$. $CR \in [0, 1]$ is the crossover probability, which determines the likelihood of inheriting elements from the mutant individual. To ensure that at least one element is inherited from the mutant individual, $j_{\mathrm{rand}}$ is a randomly chosen index that guarantees $v_{i,j}$ is selected for the corresponding $j = j_{\mathrm{rand}}$.

## C.4 Repair

Following the mutation and crossover operations, the total number of elements with value 1 may deviate from the pre-defined constraint. We should repair the count of 1s to precisely match the target patch area. Let $n$ denote the desired number of elements with value 1, and $n_c$ denote the current number of 1s in $\mathbf{m}$. If $n_c > n$, we randomly select $n_c - n$ elements with value 1 and set them to 0. If $n_c < n$, we randomly select $n - n_c$ elements with value 0 and set them to 1. This procedure ensures that the total perturbation area of the patch remains constant.

## C.5 Fitness Function

The fitness function evaluates the quality of each individual during the selection process, retaining those with higher fitness and discarding others. It is designed based on the cross-entropy loss $\mathcal{L}_{CE}$.

Specifically, pixel values from $\mathbf{r}$ are extracted and mapped to positions where $\mathbf{m}[i] = 1$, forming a perturbation $\hat{\delta}$ with dimensions $3 \times \frac{h}{4} \times \frac{w}{4}$. Using $4 \times 4$-tiling, $\mathbf{m}$ and $\hat{\delta}$ are resized to match the original image dimensions, i.e., $M$ and $\delta$ respectively. The perturbed example is then constructed using Equation (1) and fed into the target model. Subsequently, the cross-entropy loss regarding the model's output is used to assess the perturbation's impact. For untargeted attacks, the fitness function is $\mathcal{L}_{CE}(f(\hat{x}), y)$, where $y$ is the true label. For targeted attacks, it is $-\mathcal{L}_{CE}(f(\hat{x}), y_t)$, where $y_t$ is the target label.

## C.6 Selection

The selection operator in our DE algorithm determines which individuals, between the current population and the newly generated trial vectors, will survive to form the population for the next generation. This process employs a one-to-one greedy selection strategy based on fitness values.

Let $P = \{p_1, \ldots, p_N\}$ be the current population at generation $t$, where each individual $p_i = (\mathbf{m}_i, \mathbf{r}_i)$ consists of an encoded mask and content. Let $U = \{u_1, \ldots, u_N\}$ be the population of trial vectors generated through mutation and crossover from $P$, where each $u_i = (\mathbf{m}_i', \mathbf{r}_i')$. Let $F(p)$ denote the fitness function, where a higher fitness value indicates a better solution.

For each pair of corresponding individuals, their fitness values are compared. The individual with the superior fitness is selected to be part of the population for the next generation.

$$
p_i^{(t+1)} = \begin{cases} u_i^{(t)} & \text{if } F(u_i^{(t)}) > F(p_i^{(t)}) \\ p_i^{(t)} & \text{otherwise} \end{cases}
\tag{8}
$$

This elitist selection mechanism ensures that the fitness of the population is non-decreasing from one generation to the next, preserving good solutions found so far and driving the search towards more promising regions of the solution space.

## D  Effectiveness Analysis

To gain deeper insights into how IMPACT influence the model's decision-making process, we employ class activation mapping (CAM) [30] to visualize the regions the model focuses on when making predictions. Figure 7 presents CAM visualizations for several examples from the ImageNet dataset, showing the model's attention on both original images and their corresponding adversarial versions generated by IMPACT.

After applying the IMPACT-generated patches, a significant shift in the model's attention is observed in the corresponding CAMs. The model's focus is now often distracted or drawn towards the locations of our strategically placed irregular patches leading to a misclassification. This suggests that the patches introduce features that the model deems highly indicative of the incorrect class or sufficiently disruptive to confuse the features of the true class. The IMPACT method effectively manipulates the model's learned saliency. The irregular, multi-patch design appears adept at either creating new, highly salient focal points or disrupting the existing saliency map of the true object. The distributed nature of the multi-patches allows for influencing model attention at several locations simultaneously, potentially being more effective than a single, larger patch in confusing the model's global understanding of the scene.

The CAM results provide qualitative evidence for the effectiveness of our IMPACT. The patches do not need to be large or overtly cover the main object to be effective; instead, their carefully optimized content and placement, facilitated by our DE-based joint optimization and irregular shape generation, are sufficient to significantly alter the model's feature interpretation and subsequent attention, leading to successful attacks.

| Original Image | Original Image CAM | Adversarial Image | Adversarial Image CAM |

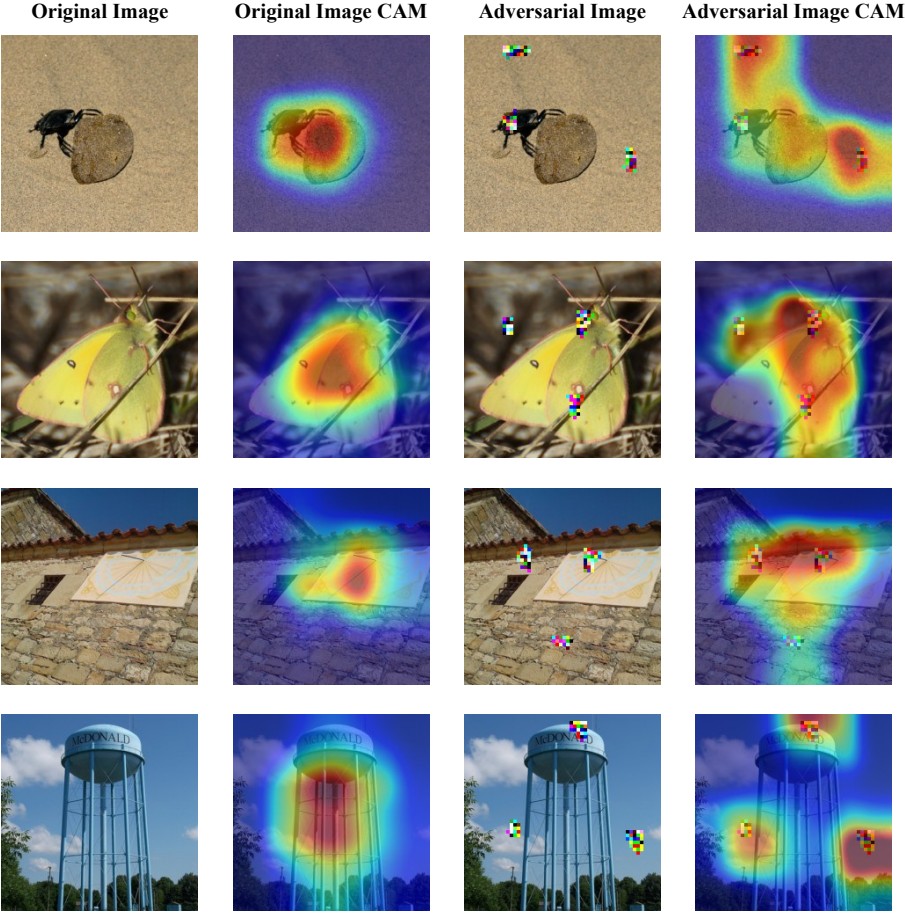

Figure 7: The figure is organized into four columns per example: (1) the original image, (2) the CAM for the original image correctly classified by the ResNet50 model, (3) the adversarial image generated by IMPACT, and (4) the CAM for the adversarial image, now misclassified by the model.

## E   Experimental Details

### E.1   Parameter Settings

**Black-box Comparison**: We evaluate IMPACT under different perturbation areas, with parameters $n = 32, 64$ controlling perturbation areas of 1% and 2%, respectively. The remaining parameters of IMPACT are set as follows: $k = 3$, $N = 50$, $T_d = 150$, and $T_e = 2500$. For the comparison methods, since these methods also support the ImageNet dataset, we set the parameters to their recommended values as suggested in the respective papers.

**White-box Comparison**: The parameters of IMPACT are $N = 50$, $T_d = 150$, and $T_e = 2500$. For a perturbation area of 0.5%, we set $n = 16$ and $k = 1$. For a perturbation area of 1%, we set $n = 32$ and $k = 2$. For a perturbation area of 1.5%, we set $n = 48$ and $k = 3$. To ensure fairness in comparison, we align the patch number for Patch-Fool with that of IMPACT at different perturbation areas. Additionally, when optimizing patch content, both IMPACT and Patch-Fool use 250 iterations of PGD.

### E.2   Evaluation Metrics

Here, we present the formulas for ASR and AQ. For the set of input images $N_{\text{clean}}$, which the model correctly classifies without any attack, the untargeted ASR is defined as:

$$\text{ASR}_{\text{untargeted}} = \frac{|N_{\text{misclassified}}|}{|N_{\text{clean}}|} \times 100\%$$

where $N_{\text{misclassified}}$ is the subset of $N_{\text{clean}}$ that is misclassified after applying adversarial patches. $|\cdot|$ is the size of the set.

The targeted ASR is defined as:

$$\text{ASR}_{\text{targeted}} = \frac{|N_{\text{targeted}}|}{|N_{\text{clean}}|} \times 100\%$$

where $N_{\text{targeted}}$ is the subset of $N_{\text{clean}}$ where adversarial patches successfully causes the model to classify the images into a specific, pre-defined target class.

AQ measures the average number of queries required to successfully craft adversarial patches. For the set of input images $N_{\text{clean}}$, it is defined as:

$$\text{AQ} = \frac{\sum_{i \in N_{\text{clean}}} q_i}{|N_{\text{clean}}|}$$

where $q_i$ is the number of queries required to successfully attack the $i$-th image. For unsuccessful attacks, $q_i$ is set to the query limit $Q_{max}$.

### E.3 Sensitivity of Random Seeds

Given the stochastic elements inherent in our IMPACT framework, we investigated the method's sensitivity to random seed. We performed 10 independent runs for untargeted attacks with both 1% and 2% total perturbation areas against the ResNet50 model, each run utilizing a different random seed. The total query budget was fixed at 5000. Figures 8 illustrates the distribution of the ASR and AQ across seeds.

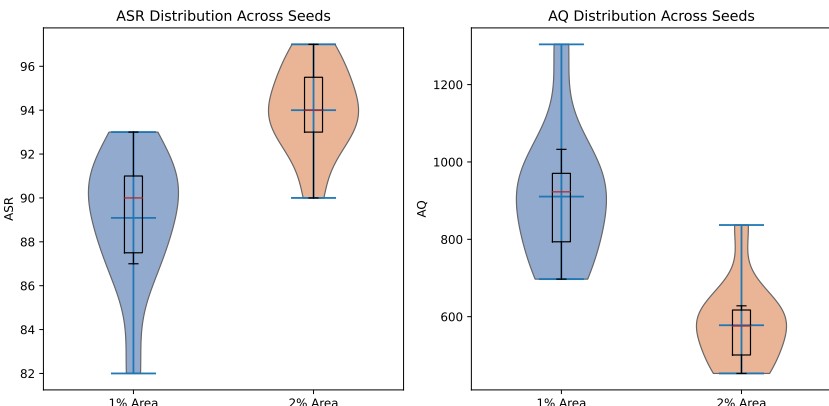

Figure 8: Violin plots of ASR and AQ across 10 random seeds for two patch budgets (1 % vs. 2 % of patch area). The shaded violins depict the full distribution density, the black boxes show the interquartile range with median lines, and the orange ticks mark the mean. Allowing a larger patch area (2 %) yields higher and more consistent ASR while reducing the number of queries needed.

The violin plots for both ASR and AQ demonstrate that our IMPACT method exhibits good robustness to initialization stochasticity. While some variation is expected in heuristic search algorithms, the results are largely consistent, especially at a 2% perturbation area where high success rates are reliably achieved with stable query efficiency. This indicates that IMPACT's performance is not overly sensitive to the specific random seed chosen, making it a reliable method for generating adversarial patches.

### E.4 Comparison of White-Box Attack Methods

We designed white-box experiment to evaluate the contribution of our patch mask generation strategy. To ensure a fair comparison, we create a white-box version of our IMPACT method, referred to

as IMPACT-W. Specifically, after obtaining the multi-patch mask $M$ from DE, we use Projected Gradient Descent (PGD) [23] to optimize the patch content. Patch-Fool [14] is a state-of-the-art methods, and it has shown superior performance compared to other white box methods, such as LOAP [28], DPA [5]. Patch-Fool offers multiple strategies for selecting patch locations. We focus on two: saliency-based selection (Patch-Fool-S) and random selection (Patch-Fool-R). Table 4 presents the comparison with Patch-Fool, based on consistent perturbation areas and patch counts: 0.5% (1 patch), 1% (2 patches), and 1.5% (3 patches). The parameter settings of IMPACT-W and Patch-Fool are in Appendix E.1. Results show our method outperforms Patch-Fool, highlighting the advantage of optimizing patch shapes and demonstrating the superiority of our patch mask generation approach.

Table 4: Performance comparison of white-box methods.

| Model | Method | ASR | | |
|---|---|---|---|---|
| | | 0.5% | 1% | 1.5% |
| ResNet50 | IMPACT-W | 80.50 | 95.40 | 98.50 |
| | Patch-Fool-S | 76.00 | 93.40 | 97.40 |
| | Patch-Fool-R | 68.40 | 90.40 | 96.80 |

## E.5  Evaluation on More Datasets

To further validate the generalization capability of IMPACT, we extended our experiments beyond ImageNet to include CIFAR-10 and CIFAR-100 using ResNet50 as the backbone model. Table 5 summarizes the results under both untargeted and targeted settings. Given the lower resolution of CIFAR images ($32 \times 32$), we adjusted the encoding to employ a $1 \times 1$ tiling configuration. This modification ensures maximum granularity while maintaining a manageable search space, highlighting the flexibility of our encoding design.

Table 5: Performance on CIFAR-10 and CIFAR-100 datasets using ResNet50, with 5000 query budget and 5% perturbation area.

| Dataset | Attack Type | Method | AQ $\downarrow$ | ASR (%) $\uparrow$ |
|---|---|---|---|---|
| CIFAR-10 | Untargeted | Patch-RS | 156.78 | 97.8 |
| | | IMPACT | **113.98** | **99.2** |
| | Targeted | Patch-RS | 2155.47 | 85.3 |
| | | IMPACT | **1634.36** | **92.7** |
| CIFAR-100 | Untargeted | Patch-RS | 98.52 | 98.1 |
| | | IMPACT | **72.81** | **99.3** |
| | Targeted | Patch-RS | 2310.80 | 82.5 |
| | | IMPACT | **1556.17** | **90.1** |

As shown in Table 5, IMPACT achieves near-perfect ASRs exceeding 99% in untargeted attacks across both CIFAR-10 and CIFAR-100, outperforming the strong baseline Patch-RS. Moreover, even in the more challenging targeted setting, IMPACT maintains ASRs above 90%, significantly reducing the query cost compared to Patch-RS. These results demonstrate that IMPACT's effectiveness is not limited to a particular data distribution such as ImageNet, but rather generalizes well to datasets with distinct statistical and visual characteristics.

## E.6  Evaluation on Defense Models

To further assess the robustness of our IMPACT method, we evaluated its performance against several defense mechanisms. Similar to reference [43], we evaluate the effectiveness of our IMPACT method against adversarially trained models [29] and the PatchGuard defense [44].

Table 6 summarizes the ASR and AQ of our IMPACT method when attacking these defended models, compared to its performance against a standard, non-defended ResNet50 model. The results demonstrate that while defenses can impact performance, our IMPACT method exhibits considerable resilience, particularly against the tested adversarially trained models. Against the specialized

PatchGuard defense, IMPACT's success rate, while reduced, remains significant, indicating its potential to overcome defenses specifically designed for patch attacks. The increased query cost in this scenario highlights the added difficulty imposed by such a defense.

Overall, these findings underscore the strength of IMPACT as a black-box patch attack. Its ability to jointly optimize diverse patch characteristics allows it to remain effective even when faced with common and specialized defense strategies, motivating further research into more comprehensive defense mechanisms.

Table 6: ASR and AQ of IMPACT against standard and defended models (Black-box, untargeted, 5000 queries, 2% area, 3 patches).

| Target Model | ASR (%) | AQ |
|---|---|---|
| ResNet50 (No Defense) | 94.2 | 676 |
| AT-ResNet50-L2 ($\epsilon = 3.0$) | 92.2 | 520.6 |
| AT-ResNet50-Linf ($\epsilon = 4/255$) | 93.8 | 508.1 |
| bagnet17 with PatchGuard | 83.4 | 1920.6 |

### E.7  Physical-World Evaluation

To assess the real-world applicability of IMPACT, we conducted physical-world experiments following Wang et al. [39]. Specifically, we randomly selected 100 images with adversarial patches and printed each on a 10 cm × 10 cm white paper. Using an iPhone 15, we photographed each printed image from different distances (10 cm, 15 cm, and 20 cm) and viewing angles (0°, 15°, and 30°). The captured photos were then resized to 224 × 224 pixels for input to the ResNet50 model. All experiments were conducted under the targeted attack setting, where the goal was to force the model to predict a specific incorrect class.

Table 7: Physical-world performance of IMPACT. The targeted ASR (%) is reported across varying distances and viewing angles for ResNet50 model. "Digital ASR" denotes the corresponding performance in the digital domain for the same 100 images.

| Model | Digital ASR | Angle | ASR@10cm | ASR@15cm | ASR@20cm |
|---|---|---|---|---|---|
| ResNet50 | 53.0 | 0° | 41.0 | 37.0 | 32.0 |
| | | 15° | 35.0 | 31.0 | 26.0 |
| | | 30° | 28.0 | 25.0 | 21.0 |

As shown in Table 7, IMPACT remains effective in real-world settings. Even under challenging conditions (20 cm distance and 30° viewing angle), it achieves a 21.0% targeted ASR. The observed degradation from the digital to the physical domain is expected, as variations in distance and viewing angle inevitably diminish patch detail in captured images. Despite this, IMPACT retains meaningful attack performance, demonstrating its practical feasibility.

### E.8  Consumption Time Analysis

To assess the computational overhead of IMPACT, we follow Williams et al. [43], measuring the average time required to successfully complete an attack on a single image. The experiments were conducted on a single NVIDIA RTX 4090 GPU under the untargeted attack setting against the ResNet50 model, with a query budget of 5000 and a perturbation area of 2%. The results are summarized in Table 8.

Table 8: Comparison of computational time under the untargeted attack setting on ResNet50.

| Method | ASR (%) ↑ | AQ ↓ | Runtime (s) ↓ |
|---|---|---|---|
| IMPACT | **94.2** | **676** | 29.16 |
| Patch-RS | 89.8 | 982 | **1.78** |

As shown in Table 8, IMPACT requires more time per attack (29.16 s) than Patch-RS (1.78 s), primarily because it employs a population-based optimization algorithm that evaluates multiple

candidate solutions per generation. In contrast, Patch-RS relies on a simpler random sampling strategy with a lower per-iteration computational cost.

Despite the higher runtime, IMPACT significantly outperforms Patch-RS in the metrics that matter most in black-box attack scenarios. Specifically, it achieves a higher attack success rate (**94.2% vs. 89.8%**) and requires substantially fewer model queries (**676 vs. 982** on average). In real-world applications, each query can be expensive and increases the risk of detection, making query efficiency far more critical than raw computational time. Thus, while IMPACT incurs a higher runtime than Patch-RS, this is justified by its substantially better attack quality and query efficiency, and the overall runtime remains practical for real-world deployment.

### E.9 Ablation Study on Random Aggregation

Our Random Aggregation algorithm incorporates stochasticity at several key junctures to foster diversity in the generated irregular patch shapes. Specifically, these include: (1) Random aggregation center. The center for aggregation within each K-Means cluster is chosen randomly from the cluster's members. (2) Random target in neighborhood. When an element is being moved, the target towards which it moves is randomly selected from the neighborhood of the aforementioned aggregation center. (3) Random movement direction. If multiple directions (e.g., horizontal and vertical) would reduce an element's distance to its target, one is chosen randomly. To understand contributions of these stochastic elements, we conducted an ablation study in Table 9.

Table 9: Ablation study on stochastic components within random aggregation.

| Aggregation Variant | ASR (%) | AQ |
| --- | --- | --- |
| Full random | 94.2 | 676 |
| No random aggregation center | 89.5 | 782 |
| No random target in neighborhood | 92.1 | 727 |
| No random movement direction | 92.4 | 684 |

For no random aggregation center, instead of randomly selecting an aggregation center from within the cluster members, we deterministically use the centroid calculated by the K-Means algorithm as the aggregation center for each cluster. All other stochastic elements remain active. For no random target in neighborhood, elements are moved directly towards the aggregation center itself, rather than to a randomly chosen point within its neighborhood. For no random movement direction, the algorithm always attempts a horizontal move first, followed by a vertical move.

This ablation study highlights the importance of the stochastic elements within our Random Aggregation algorithm. The random selection of the aggregation center appears to be the most impactful, significantly contributing to both attack success and efficiency. Randomness in selecting the target within the center's neighborhood also provides a noticeable benefit. While randomizing the movement direction has a smaller effect on the primary metrics, retaining all three sources of randomness likely contributes to the overall diversity and robustness of the patch generation process. These findings justify our design choices to incorporate these levels of stochasticity to enhance the exploration of diverse and effective irregular patch shapes.

### E.10 Parameter Experiment Details

In our method, the number of queries is influenced by three parameters: $N$, $T_d$, and $T_e$. The total number of queries is calculated as $Q = N \times T_d + T_e$. To assess the impact of these parameters on optimization performance, we design experiments focusing on $N$ and $T_d$. We set a total query budget of 10,000 while keeping $n$, $k$, and $T_e$ constant. This approach resulted in various combinations of $N$ and $T_d$. Table 10 illustrates the effect of these different combinations. It can be observed that the highest ASR 95.20% is achieved when $N = 50$ and $T_d = 150$. Under a fixed query budget, prioritizing a higher number of iterations over a larger population size leads to better optimization and more effective attacks.

We further investigate the sensitivity of the DE phase to its two core hyperparameters: the mutation factor $F$ and the crossover probability $CR$. All experiments below use only Phase 1 optimization with $N = 50$, $T_d = 100$, and a fixed total query budget of $Q = 5\,000$. We sweep the two core DE

Table 10: The impact of different parameter combinations.

| $n$ | $k$ | $T_e$ | $N$ | $T_d$ | ASR(%) |
|-----|-----|-------|-----|-------|--------|
| 64 | 3 | 2500 | 25 | 300 | 94.60 |
| | | | 50 | 150 | 95.20 |
| | | | 100 | 75 | 94.00 |
| | | | 150 | 50 | 93.80 |

hyperparameters: mutation factor $F$ and crossover rate $CR$, while holding the total patch area at 2% and the number of patches $k = 3$ constant. Specifically, when evaluating $CR$, we fix $F = 2$, and when evaluating $F$, we fix $CR = 0.8$. The results are reported in Table 11 and Table 12, respectively.

<table>
<tr><td colspan="3">Table 11: Effect of crossover rate $CR$.</td><td colspan="3">Table 12: Effect of mutation factor $F$.</td></tr>
<tr><td>$CR$</td><td>ASR (%)</td><td>AQ</td><td>$F$</td><td>ASR (%)</td><td>AQ</td></tr>
<tr><td>0.2</td><td>90.4</td><td>835</td><td>0.5</td><td>85.4</td><td>1058</td></tr>
<tr><td>0.4</td><td>88.2</td><td>840</td><td>1.0</td><td>87.8</td><td>835</td></tr>
<tr><td>0.6</td><td>89.2</td><td>785</td><td>2.0</td><td>90.8</td><td>748</td></tr>
<tr><td>0.8</td><td>90.8</td><td>748</td><td>3.0</td><td>89.6</td><td>718</td></tr>
<tr><td>1.0</td><td>89.4</td><td>727</td><td>4.0</td><td>89.2</td><td>743</td></tr>
</table>

Our analysis of these hyperparameter sweeps reveals that IMPACT's performance is relatively robust to variations in the mutation factor $F$ and crossover rate $CR$ within typical ranges. However, to optimize the balance between success rate and query efficiency, we identified $CR = 0.8$ and $F = 2.0$ as offering a marginally superior trade-off. Consequently, these values are adopted as defaults in other reported experiments.

