# OpenReview forum: "IMPACT: Irregular Multi-Patch Adversarial Composition Based on Two‑Phase Optimization"
_NeurIPS.cc/2025/Conference — NeurIPS 2025 poster_

### Official Review · Reviewer_Tn9c · 2025-06-24

**Clarity:** 4
**Significance:** 3
**Originality:** 3
**Rating:** 4
**Confidence:** 3

**Summary:**

This paper introduces IMPACT, a novel adversarial patch attack framework that jointly optimizes all critical patch factors—shape, location, number, and content—through a two-phase, gradient-free optimization process. The authors propose a dimensionality-reduced encoding scheme and a random aggregation algorithm to produce irregular but physically feasible patches. The method demonstrates strong performance across multiple black-box attack settings and outperforms several state-of-the-art baselines in both untargeted and targeted attacks.

**Questions:**

Please refer to the Weaknesses section and address each point.

**Ethical Concerns:**

["NO or VERY MINOR ethics concerns only"]

**Final Justification:**

The paper provides solid experimental contributions, including physical validation, but novelty and theoretical depth are limited. Concerns on assumptions and runtime remain, so I keep Borderline Accept.

**Limitations:**

Even the proposed method belongs to the black-box AA category, the method assumes access to model confidence scores. This limits its applicability to real-world systems that only expose top-1 labels. Decision-based adaptation is mentioned as future work but not explored.

**Quality:**

3

**Strengths And Weaknesses:**

Strengths:
1. The paper tackles the underexplored problem of joint optimization over all patch attributes, including irregular geometry and multiple patch composition—areas often overlooked or simplified in prior works.
2. The dimensionality-reduction strategy via 4×4 tiling and the stochastic patch aggregation allow for tractable search while producing diverse, physically printable adversarial examples.
3. The paper evaluates the contribution of each phase, different optimizers, and tile granularity, supporting the design choices with data.

Weaknesses:
1. The method assumes access to model confidence scores. This limits its applicability to real-world systems that only expose top-1 labels. Decision-based adaptation is mentioned as future work but not explored.
2. While effective empirically, the method lacks any theoretical insight into convergence behavior, success conditions, or the impact of irregular geometry on attack transferability.

---

> ### Author Rebuttal · Authors · 2025-07-31
>
> Thank you for your valuable comments.
>
> ### **W1. Decision-Based Adaptation:**
>
> We sincerely thank you for this important question. We agree that the decision-based setting is equally important and poses stricter constraints. However, the score-based setting **remains highly relevant in practice** [1-3], as many real-world systems (e.g., commercial APIs) expose confidence scores. Our focus on this setting reflects a deliberate effort to tackle the under-explored challenge of jointly optimizing multiple entangled patch factors under non-differentiable, black-box conditions. We believe this problem is both technically rich and practically impactful.
>
> Meanwhile, extending to decision-based attacks requires fundamentally different query strategies and objective formulations, as only the top-1 predicted label is available and class probabilities are inaccessible. This setting severely limits gradient-free optimization, especially when optimizing multiple entangled patch factors, as there is no feedback on directionality or partial improvement. Adapting IMPACT to this restrictive scenario would necessitate new strategies, such as label-guided boundary estimation or discrete search refinements, which go beyond the current scope but represent a **promising direction for future work**.
>
> In the revised paper, we will expand the discussion of future work to explicitly include decision-based attacks. We will highlight the importance of this setting and outline how IMPACT could be extended to cope with the associated challenges. By positioning our work as a **foundation for such extensions**, we aim to emphasize the broader applicability and long-term potential of the IMPACT framework across diverse black-box threat models.
>
> ### **W2. Theoretical Insight:**
>
> We acknowledge that theoretical analysis would strengthen the paper. However, analyzing population-based algorithms like DE on high-dimensional, non-convex black-box objectives **remains an open challenge**. To this end, we focus on conducting a **comprehensive and in-depth empirical evaluation**, including detailed CAM visualizations and ablation studies, to support our design choices and hypotheses.
>
> This empirical approach follows **a well-established practice in adversarial machine learning**, where many impactful papers published in top-tier conferences and journals [1-4] have similarly relied on **strong empirical validation in lieu of theoretical guarantees**.
>
> Our empirical results consistently demonstrate the effectiveness of irregular multi-patch attacks that are optimized jointly across shape, location, number, and content. This unified optimization framework enables more effective perturbations that significantly degrade model performance. The optimized patches can also disrupt learned saliency patterns by directly interfering with the model’s internal mechanisms for identifying and attending to salient input regions (i.e., saliency or attention attribution). Hence, our empirical study comprehensively and strongly supports the major claim of this paper that irregular, multi-patch configurations are particularly effective in **diffusing attention and triggering distributed misclassification**.
>
> **We hope our responses have fully addressed your concerns. We would deeply appreciate it if you would consider our clarifications in your final evaluation. Please feel free to let us know if there are any further questions. We would be happy to provide additional responses.**
>
>
> **Reference:**
>
> [1] Croce et al. Sparse-RS: A Versatile Framework for Query-Efficient Sparse Black-Box Adversarial Attacks. AAAI 2022
>
> [2] Williams et al. CamoPatch: An Evolutionary Strategy for Generating Camouflaged Adversarial Patches. NeurIPS 2023
>
> [3] Yang et al. PatchAttack: A Black-Box Texture-Based Attack with Reinforcement Learning. ECCV 2020
>
> [4] Chen et al. Query-Efficient Decision-Based Black-Box Patch Attack. TIFS 2023

---

> > ### Comment · Reviewer_Tn9c · 2025-08-04
> > **Thank you for the thoughtful response.**
> >
> > Thank you for the thoughtful response. I appreciate the clear articulation of potential theoretical directions and the rationale behind focusing on empirical validation in this version. The additions to the revised manuscript help clarify the scope and intention of the work. While I still believe some minimal theoretical modeling could further elevate the contribution, I understand and accept the authors’ decision to retain their empirical framing. Looking forward to future work that builds on this promising foundation.

---

> > > ### Author Response · Authors · 2025-08-06
> > >
> > > We sincerely thank you for your encouraging and thoughtful feedback. We greatly appreciate your understanding of our decision to focus on empirical validation in this version, and we are glad to hear that our responses have helped clarify the intent of our work.
> > >
> > > Your thoughtful suggestion motivates us to pursue a more formal theoretical analysis in subsequent work, and we look forward to building on this foundation. Thank you again for your support and insightful comments.

---

### Official Review · Reviewer_eyLv · 2025-06-28

**Clarity:** 3
**Significance:** 3
**Originality:** 3
**Rating:** 5
**Confidence:** 4

**Summary:**

This work mainly focuses on how to generate adversarial patches to improve the success rate of adversarial attacks. The authors proposed a flexible adversarial attack framework based two phases optimization and introduced a new aggregation algorithm explicitly designed to produce contiguous, irregular patches. The experiments have shown the effectiveness of the method.

**Questions:**

1. Could the authors provide information about the differences in the calculation of consumption time between IMPACT and the previous method?
2. I am confused about some of the function names in Algorithm 1, such as PopInit, Fitness, Mutation and so on. Could the author provide some more precise formulas in the main paper so that the reader can make sense of them.
3. The results of IMPACT on some defense models in Table 5 are very good, how about the results on diffusion-based defense methods such as Diffpure[1]?
4. Have the authors printed patches attacking some models in the physical world, as mentioned in line 73?

[1] Nie et al. Diffusion Models for Adversarial Purification. ICML 2022.

**Ethical Concerns:**

["NO or VERY MINOR ethics concerns only"]

**Final Justification:**

The author has addressed my concerns in the rebuttal, and I am willing to raise my rating to “accept.”

**Limitations:**

No discussion of the time consumption of the proposed method compared to previous methods, and evaluation with more advanced defense models.

**Paper Formatting Concerns:**

No formatting concerns

**Quality:**

3

**Strengths And Weaknesses:**

Paper strengths:
1. Unlike the independent optimization of patch and position, this paper presents a novel joint optimization strategy based on evolutionary algorithms.
2. They develop a new dimensionality reduction method that significantly reduces the encoding length while addressing limitations in existing approaches. Figure 2 gives a clear presentation of this approach.
3. The experimental results demonstrate the effectiveness of the current method.

Paper Weakness:
1. The authors did not evaluate the time consumed for joint optimization using evolutionary algorithms compared to some previous work.
2. There are some function names in Algorithm 1 that the authors do not explain in the main paper.

---

> ### Author Rebuttal · Authors · 2025-07-31
>
> Thank you for your valuable comments.
>
> ### **Q1. Consumption Time:**
> We thank the reviewer for questioning the **computational time**. To assess the computational overhead, we follow Williams et al. [1], measuring the average time required to successfully complete an attack on a single image. The experiments were conducted on a single NVIDIA RTX 4090 GPU under the untargeted attack setting against ResNet50 (5000 query budget and 2\% perturbation area). The results are presented in Table 1.
>
> | Method   | ASR (%)         | AQ              | Runtime (s)     |
> | :------- | :-------------- | :-------------- | :-------------- |
> | IMPACT   | **94.2**        | **676**         | 29.16       |
> | Patch-RS | 89.8            | 982             | **1.78**      |
>
> *Table 1: Comparison of Performance and Computational Time
> (Untargeted Attack on ResNet50, 5000 Query Budget, 2% Perturbation Area).*
>
> As shown in Table 1, IMPACT takes more time per attack (29.16s) than Patch-RS (1.78s), primarily because it employs a **population-based** optimization algorithm (Differential Evolution) that **evaluates multiple candidate solutions per generation**. In contrast, Patch-RS relies on a simpler random sampling strategy with lower per-iteration cost.
>
> Despite the longer runtime, IMPACT significantly outperforms Patch-RS in the metrics that matter most in black-box attack scenarios. Specifically, it achieves a significantly **higher attack success rate** (ASR: 94.2% vs. 89.8%) and requires substantially **fewer model queries** (676 vs. 982 on average), which is critical in real-world applications where each query is expensive and increases the risk of detection. In fact, according to our knowledge, **query count**, rather than computational time, is widely regarded as the primary bottleneck in black-box attacks. Many state-of-the-art methods [2-5] focus on minimizing query cost since queries dominate both efficiency and stealth.
>
> Finally, we note that IMPACT remains computationally practical: its average runtime (29.16s) is **more than an order of magnitude faster** than the 440s reported for CamoPatch [1], a state-of-the-art patch-based method.
>
> Thus, while IMPACT incurs a higher runtime than Patch-RS, this is **justified by substantially better attack quality and efficiency**. In view of the newly reported results, we believe the runtime of IMPACT remains practical, particularly faster than other complex methods like CamoPatch.
>
> ### **Q2. Function Explanation in Algorithm 1:**
>
> We thank the reviewer. In the revised paper, we will add a concise summary of each function (PopInit, Mutation, Crossover, Aggregation, Fitness, Selection, SelectBest) immediately following Algorithm 1, with references to relevant appendices.
>
> To address the reviewer's concern, we provide a concise summary of these functions here.
>
> - **PopInit** (Initialization, detail in the Appendix B.1): This function initializes a population $P_0$ of $N$ individuals to ensure diverse starting points for the search process. Each individual is structured according to our dimensionality reduction encoding scheme, consisting of two components: a binary encoded mask $\mathbf{m}$ and a continuous encoded content $\mathbf{r}$.
>
> - **Mutation** (Details in the Appendix B.2): This core DE function creates a mutant vector $v_i$ for each individual $p_i$ by differential combination of other population members. This process introduces new variations into the search.
>
> - **Crossover** (Details in the Appendix B.3): This function mixes the components of the parent individual $p_i$ and its corresponding mutant vector $v_i$ to create a new trial individual $u_i$. This enhances population diversity.
>
> - **Aggregation** (Details in the Section 3.2.2 Random Aggregation): This is a component of our framework. It takes the sparse binary mask $m$ from a trial individual and transforms its scattered active elements into $k$ physically contiguous, irregular patches.
>
> - **Fitness** (Details in the Appendix B.5): This function evaluates the quality of a trial individual. It reconstructs the full patch, applies it to the image, and calculates the model's cross-entropy loss.
>
> - **Selection** (Details in the Appendix B.6): This function compares the fitness of the trial individual $u_i$ with the parent $p_i$. The one with the better fitness score survives into the next generation's population.
>
> - **SelectBest**: This is a simple function that iterates through the final population and returns the single individual with the best fitness score found during the optimization.
>
> ### **Q3. Performance on Diffusion-Based Defense Methods:**
>
> We thank the reviewer for raising this question regarding diffusion-based defense methods like DiffPure.
>
> We agree that testing against diffusion-based defenses is a **crucial next step**. In our initial experimental design, following Williams et al. [1], we focused on more established defense benchmarks, such as adversarially trained models and PatchGuard.
>
> While we have not yet conducted these specific experiments, we can offer an **analysis** of how IMPACT might interact with a defense like DiffPure. Diffusion-based defenses work by purifying the input image. They use a diffusion model to effectively remove high-frequency adversarial perturbations. This process fundamentally alters the gradient landscape, making it very difficult for gradient-based attacks to succeed.
>
> IMPACT generates **irregular**, **multi-patch** attacks which are **structurally very different** from the subtle, pixel-level noise that diffusion models are primarily trained to remove. It is plausible that our patches might partially survive the purification process, or that they could be optimized to create features that mislead the model even after purification.
>
> The reviewer's question highlights an exciting avenue for future research. We are highly motivated to rigorously evaluate IMPACT against a suite of diffusion-based defenses in our future work. It is also worthwhile to emphasize that **the design and development of IMPACT itself represents a substantial technical contribution**. IMPACT introduces a novel encoding scheme and a joint optimization framework capable of evolving shape, location, and content of patches under strict black-box constraints. This unified approach addresses key limitations of prior methods and advances the state-of-the-art in patch-based attacks, independent of downstream defense evaluations.
>
> ### **Q4. Performance in the Physical World:**
>
> We sincerely thank the reviewer for this important question. Demonstrating the physical-world applicability of our method is indeed a critical part of our evaluation.
>
> To validate the physical-world effectiveness of our method, we conducted real-world experiments following Wang et al. [6]. Specifically, we randomly selected 100 images with adversarial patches and printed each on a 10cm × 10cm white paper. Then, using an iPhone 15, we photographed each printed image at various distances and angles. All photos were then resized to 224 × 224 pixels for ResNet50 model input. Table 2 reports the resulting ASR across the different shooting distances and viewing angles. Note that all these results are for **targeted attacks**. This means the goal was not simply to cause a misclassification, but to force the model to predict a specific, pre-determined incorrect class.
>
>
> | Model    | Digital ASR   | Angle| ASR at 10cm  | ASR at 15cm  | ASR at 20cm  |
> |----------|-----------|------|-------|-------|-------|
> | ResNet50 | 53.0      | 0°   | 41.0    | 37.0    | 32.0    |
> |          |           | 15°  | 35.0    | 31.0    | 26.0    |
> |          |           | 30°  | 28.0    | 25.0    | 21.0    |
>
> *Table 2: Physical-world performance of IMPACT under different conditions. The targeted attack success rates ASR (%) is reported across varying distances and viewing angles for ResNet50 model. The "Digital ASR" indicates the attack success rate of the same 100 selected images in the digital domain.*
>
> These results confirm that our irregular patches **remain effective when physically deployed**. Even under the challenging condition (20cm distance and a 30° viewing angle), IMPACT still achieves a 21.0\% ASR. It is worth noting that even a strong baseline like Patch-RS only achieves a 20.0\% ASR in the digital-world targeted attack setting (Table 1 in our main paper). In the physical world, varying distances and angles inevitably reduce the patch's detail in the captured image. Consequently, a degradation in performance when moving from the digital domain to the physical world is an expected phenomenon. **Despite these challenges, our method demonstrates resilience.** This validates the practical applicability and physical feasibility of IMPACT.
>
> In the final version of the paper, we will include the analysis of these real-world experiments. All physical-world images will be released upon acceptance of the paper.
>
> **We hope that our clarifications have resolved the issues you raised. We believe the manuscript is significantly stronger thanks to your feedback, and we would sincerely appreciate it if you would take our responses into account in your final assessment. We remain available for any further questions you may have.**
>
> **Reference:**
>
> [1] Williams et al. CamoPatch: An Evolutionary Strategy for Generating Camouflaged Adversarial Patches. NeurIPS 2023
>
> [2] Chen et al. Query-Efficient Decision-Based Black-Box Patch Attack. TIFS 2023
>
> [3] Croce et al. Sparse-RS: A Versatile Framework for Query-Efficient Sparse Black-Box Adversarial Attacks. AAAI 2022
>
> [4] Fu et al. PATCH-FOOL: Are Vision Transformers Always Robust Against Adversarial Perturbations? ICLR 2022
>
> [5] Ran et al. Cross-Shaped Adversarial Patch Attack. IEEE Transactions on Circuits and Systems for Video Technology 2024
>
> [6] Wang et al. Breaking Barriers in Physical-World Adversarial Examples: Improving Robustness and Transferability via Robust Feature. AAAI 2025

---

> > ### Comment · Reviewer_eyLv · 2025-08-05
> >
> > Thank you for the detailed rebuttal and for addressing my concerns. I also appreciate you providing the new experiments on running time and the performance in the physical world. These additions will strengthen the paper's overall scope and impact. Given these comprehensive responses and the authors' commitment to implementing these changes, I am confident that the revised paper will be much stronger. My concerns have been adequately addressed.

---

> > > ### Author Response · Authors · 2025-08-06
> > >
> > > Thank you very much for your positive and encouraging feedback. We are delighted to hear that our rebuttal have adequately addressed your concerns. We confirm that the final paper will be revised accordingly to include all these additional results and discussions. We truly appreciate your time and constructive guidance throughout this process.

---

### Official Review · Reviewer_sVYk · 2025-06-30

**Clarity:** 3
**Significance:** 2
**Originality:** 2
**Rating:** 4
**Confidence:** 2

**Summary:**

This paper introduces IMPACT, a novel framework for generating adversarial patches through the joint optimization of multiple critical factors (shape, location, number, and content). Addressing limitations of prior works that optimize these factors independently, IMPACT features three key innovations: (1) A dimensionality reduction encoding scheme that compresses the high-dimensional search space while preserving expressiveness; (2) A random aggregation algorithm that merges scattered modifications into physically realizable, contiguous irregular patches; and (3) A two-phase optimization approach combining differential evolution for global exploration with (1+1)-ES for local refinement. Extensive experiments on ImageNet demonstrate IMPACT's superiority over state-of-the-art methods

**Questions:**

1. Could you demonstrate the effectiveness of IMPACT-generated patches in physical-world scenarios (e.g., printed patches under varying lighting/viewing angles)?
2. Analyze how the 4×4 tiling encoding performs on higher-resolution images (e.g., 512×512). Does the method maintain query efficiency?
3. How does IMPACT compare if baselines (e.g., Patch-RS) are allowed similar query budgets for sequential optimization of shape/location/content?

**Ethical Concerns:**

["NO or VERY MINOR ethics concerns only"]

**Final Justification:**

The authors partially solved my concerns. After reading the rebuttals and the reviews from other reviewers, I would like to keep the score.

**Limitations:**

Yes.

**Paper Formatting Concerns:**

No.

**Quality:**

3

**Strengths And Weaknesses:**

Strengths
1. Comprehensive experiments on multiple architectures (ResNet50, VGG16, ViT-B) with clear benchmarks against SOTA methods。
2. Detailed ablation studies validating design choices (e.g., Table 2 shows 94.2% vs 90.2% ASR for full IMPACT vs DE-only).
3. The two-phase optimization framework is well-motivated and clearly explained (Algorithm 1, Section 3.2–3.3).

Weaknesses
1. While IMPACT outperforms baselines, the absolute gains in ASR are modest (e.g., +2.8% for ResNet50)
2. Comparison to white-box methods (Table 4) is less compelling since IMPACT-W uses gradient-based refinement.

---

> ### Author Rebuttal · Authors · 2025-07-31
>
> Thank you for your valuable comments.
>
> ### **W1. Modest Performance:**
>
> We thank the reviewer for this comment. While the ASR gain may appear modest in isolation in some untargeted settings, it is accompanied by a **substantial reduction in query cost** (e.g., a significant decrease of 472 queries). In practice, higher query efficiency is often as important as peak ASR in black-box setting [1].
>
> Moreover, **in targeted attacks**, where success is harder to achieve, IMPACT shows **significant gains** (e.g., 24.6% vs. 7.6% ASR on ResNet50 with 5k queries, 1% area).
>
> As far as we are aware, IMPACT is the first method to unify all patch factors under a unified optimization framework. In our humble opinion, even modest ASR improvements reflect a nontrivial **architectural advance**.
>
>
> ### **W2. White-box Comparison:**
>
> We appreciate the reviewer's observation. In the white-box comparison, both our IMPACT-W variant and the baseline Patch-Fool utilize gradient-based optimization to refine the patch content. We designed this experiment to evaluate the contribution of our **patch mask generation strategy**.
>
> IMPACT-W and Patch-Fool differ only in mask generation: IMPACT-W uses DE to optimize irregular masks, while Patch-Fool uses saliency-based or random placement with fixed shapes.
>
> In particular, both approaches use PGD to optimize content. Hence, we believe the superior performance of IMPACT-W directly demonstrates the benefit of jointly optimizing irregular mask structures in IMPACT-W.
>
> We will clarify the motivation and conclusion of this experiment in the final paper to better reflect **the advantage of IMPACT's mask optimization strategy**.
>
>
>
> ### **Q1. Performance in the Physical World:**
>
> We sincerely thank the reviewer for this important question. Demonstrating the physical-world applicability of our method is indeed a critical part of our evaluation.
>
> To validate the physical-world effectiveness of our method, we conducted real-world experiments following Wang et al. [2]. Specifically, we randomly selected 100 images with adversarial patches and printed each on a 10cm × 10cm white paper. Then, using an iPhone 15, we photographed each printed image at various distances and angles. All photos were then resized to 224 × 224 pixels for ResNet50 model input. Table 2 reports the resulting ASR across the different shooting distances and viewing angles. Note that all these results are for **targeted attacks**. This means the goal was not simply to cause a misclassification, but to force the model to predict a specific, pre-determined incorrect class.
>
>
> | Model    | Digital ASR   | Angle| ASR at 10cm  | ASR at 15cm  | ASR at 20cm  |
> |----------|-----------|------|-------|-------|-------|
> | ResNet50 | 53.0      | 0°   | 41.0    | 37.0    | 32.0    |
> |          |           | 15°  | 35.0    | 31.0    | 26.0    |
> |          |           | 30°  | 28.0    | 25.0    | 21.0    |
>
> *Table 2: Physical-world performance of IMPACT under different conditions. The targeted attack success rates ASR (%) is reported across varying distances and viewing angles for ResNet50 model. The "Digital ASR" indicates the attack success rate of the same 100 selected images in the digital domain.*
>
>
>
> These results confirm that our irregular patches **remain effective when physically deployed**. Even under the challenging condition (20cm distance and a 30° viewing angle), IMPACT still achieves a 21.0\% ASR. It is worth noting that even a strong baseline like Patch-RS only achieves a 20.0\% ASR in the digital-world targeted attack setting (Table 1 in our main paper). In the physical world, varying distances and angles inevitably reduce the patch's detail in the captured image. Consequently, a degradation in performance when moving from the digital domain to the physical world is an expected phenomenon. **Despite these challenges, our method demonstrates resilience.** This validates the practical applicability and physical feasibility of IMPACT.
>
> In the final version of the paper, we will include the analysis of these real-world experiments. All physical-world images will be released upon acceptance of the paper.
>
> ### **Q2. Scalability for Higher-Resolution Images:**
>
> We thank the reviewer for this excellent question regarding the scalability of our approach to higher-resolution images. This is a crucial aspect for practical applications, and we appreciate the opportunity to elaborate on it.
>
> Our 4 × 4 tiling strategy was **empirically optimized** for standard 224 × 224 ImageNet images, where it offers a practical balance between granularity and search space dimensionality.
>
> However, when applied to larger inputs such as 512 × 512 images, the resulting binary mask vector grows to 16,384 dimensions, over five times larger than for 224 × 224 images. This substantial increase imposes a much **heavier burden** on the DE-based optimizer, especially under fixed query budgets.
>
> As evidenced in our ablation study (Table 3), **larger search spaces may degrade query efficiency**, even when spatial expressiveness improves, especially if the 4 × 4 scheme is applied naively to images with higher resolutions.
>
> That said, we believe this limitation stems from using a fixed tiling granularity, not from our proposed framework itself. IMPACT is inherently **flexible** and supports modular adjustments to the encoding scheme. For example, a scalable variant could adapt the tile size to the input resolution (e.g., using 8 × 8 or 16 × 16 tiling for 512 × 512 images), preserving both search tractability and attack flexibility. We plan to explore such **adaptive strategies** in future work and will incorporate relevant discussions in the revised paper.
>
> ### **Q3. Compare with Sequential Optimization:**
>
> This is a insightful question that gets to the heart of the difference between our joint optimization approach and other exiting methods. We thank the reviewer for proposing this hypothetical comparison, as it allows us to highlight the fundamental advantages of IMPACT.
>
> First, we would like to clarify that the core methodology of Patch-RS is to fix the patch shape as a square and then optimize its location and content. Within this design, **'shape' is not treated as an optimizable variable**. This fundamental limitation is exactly the problem that IMPACT is designed to solve. Our framework is the first to treat shape, location, and content as **interdependent factors for joint optimization**.
>
> Secondly, even if a sequential optimization strategy were implemented, it would remain fundamentally limited compared to our joint optimization framework. This is due to the strongly **entangled nature** of all patch factors, including shape, location, number, and content. Optimizing these components in isolation, or in a fixed sequence, restricts the search to a **narrower subspace** and often results in **suboptimal configurations**.
>
> For example, selecting a location and content under the assumption of a fixed square shape may yield a solution well-suited for that specific constraint. However, when the shape is later altered, the previously selected location and content may no longer be effective, leading to suboptimal attacks or even total failure. In contrast, IMPACT jointly optimizes all patch factors within a unified encoding space, allowing it to adaptively discover **synergistic configurations** that would likely be inaccessible to sequential or factor-decoupled methods. This tight coupling is critical to achieving high adversarial success with limited query budgets.
>
>
> **We trust that this response has clarified our work and addressed your valuable points. We would be deeply grateful if you would reconsider our work in light of these clarifications. Please do not hesitate to let us know if any questions remain.**
>
> **Reference:**
>
> [1] Chen et al. Query-Efficient Decision-Based Black-Box Patch Attack. TIFS 2023
>
> [2] Wang et al. Breaking Barriers in Physical-World Adversarial Examples: Improving Robustness and Transferability via Robust Feature. AAAI 2025

---

> > ### Comment · Reviewer_sVYk · 2025-08-07
> > **Thanks for the reply**
> >
> > Thanks for the reply. It partially solved my concerns. I would like to keep the score.

---

> > > ### Author Response · Authors · 2025-08-08
> > >
> > > Thank you for your follow-up and for considering our rebuttal. We are grateful that our rebuttal was able to partially address your concerns.
> > >
> > > We are committed to incorporating your feedback to further improve the paper. If possible, could you kindly advise on how we might better address your remaining concerns? Your guidance would be greatly appreciated.
> > >
> > > Once again, thank you for your time and constructive feedback.

---

### Official Review · Reviewer_d9ew · 2025-07-01

**Clarity:** 2
**Significance:** 3
**Originality:** 2
**Rating:** 3
**Confidence:** 4

**Summary:**

This paper claims to propose a novel adversarial patch generation method that enables joint optimization of the patch’s position, number, shape, and content, without relying on gradient information. The core techniques include dimensionality-reducing encoding and a random aggregation algorithm. The proposed method shows significant improvements over baseline approaches on ImageNet.

**Questions:**

1. In line 181, what does pi refer to? Its definition is not explained.
2. In the ablation study (Table 3), the authors claim that 4×4 is the optimal tile size. However, as mentioned by the authors in line 316, smaller tile sizes should offer a higher-dimensional search space and finer granularity. So why does the attack success rate decrease with smaller tiles?
3. The paper repeatedly emphasizes the physical feasibility of the proposed method. Can experimental results be provided to demonstrate the advantages of the method in physical-world attacks?

**Ethical Concerns:**

["NO or VERY MINOR ethics concerns only"]

**Final Justification:**

In the initial review, I raised concerns about the novelty of the proposed method, the completeness of the experiments, and the details of the ablation studies. During the rebuttal phase, the authors addressed some of these concerns through explanations and additional experiments. Therefore, I raised my score to 3.

**Limitations:**

yes

**Quality:**

2

**Strengths And Weaknesses:**

Strengths:
1. The introduction of the proposed method is very detailed and easy to understand.
2. The dimensionality-reducing encoding and random aggregation algorithms are simple yet effective.
3. The experimental results are convincing.

Weaknesses:
1. The authors claim that their main contribution is proposing a method capable of jointly optimizing the position, shape, content, and number of adversarial patches. However, the method does not unify all these elements into a common feature space; instead, it simply uses two separate feature vectors to represent the various factors. This approach does not appear to differ significantly from previous methods.
2. The experiments are conducted only on ImageNet, lacking diversity in datasets.
3. The authors did not provide the full executable code—only code for key components—making it difficult to verify reproducibility.

---

> ### Author Rebuttal · Authors · 2025-07-31
>
> We thank the reviewer for their insightful and constructive comments. We have carefully addressed each concern below and will incorporate improvements into the revised paper accordingly.
>
> ### **W1. Clarify Contributions:**
>
> We appreciate the opportunity to clarify our joint optimization framework. IMPACT's key innovation lies in its ability to simultaneously optimize **all critical patch factors**, including shape, location, number, and content, using a novel unified encoding.
>
> While we employ a two-part vector encoding $(\mathbf{m},\mathbf{r})$, **these two components are not optimized in isolation**. Instead, they form **a unified representation** and are **jointly optimized in a common solution space**. The mask vector $\mathbf{m}$ implicitly defines patch shape, location, and number, while the content vector $\mathbf{r}$ specifies the corresponding pixel values. Vectors $\mathbf{m}$ and $\mathbf{r}$ compose an individual (a solution), and jointly optimized by DE.
>
> Our work addresses the problem of how to optimally craft adversarial patches by considering all critical patch factors simultaneously. In contrast, prior work has typically simplified this problem by focusing on only a subset of these factors. Technically speaking, IMPACT's holistic treatment marks a significant departure from prior methods, which typically fix specific patch parameters (e.g., shape or location) or optimize them sequentially in isolation. Such decomposed approaches fail to model the complex interdependencies among patch factors, often leading to **suboptimal adversarial effectiveness** [1-3]. In contrast, IMPACT jointly evolves both the mask vector and the content vector in a unified search space. This enables discovery of synergistic configurations that adaptively balance spatial arrangement and patch content, resulting in more effective and flexible attacks across diverse scenarios.
>
> ### **W2. Dataset:**
>
> Thank you for suggesting broader dataset evaluation. We have now conducted additional experiments on **CIFAR-10** and **CIFAR-100** using ResNet-50. The results (see Table 1 below) confirm that IMPACT **generalizes well** beyond ImageNet, maintaining high attack success rates (ASR) and strong query efficiency. Following your helpful advice, we will include these results and analysis in the final version of the paper.
>
> |Dataset|Attack Type|Method|AQ|ASR (%)|
> |-|-|-|-|-|
> |**CIFAR-10**|Untargeted|Patch-RS|156.78|97.8|
> |||**IMPACT**|**113.98**|**99.2**|
> ||Targeted|Patch-RS|2155.47|85.3|
> |||**IMPACT**|**1634.36**|**92.7**|
> |**CIFAR-100**|Untargeted|Patch-RS|98.52|98.1|
> |||**IMPACT**|**72.81**|**99.3**|
> ||Targeted|Patch-RS|2310.80|82.5|
> |||**IMPACT**|**1556.17**|**90.1**|
>
> *Table 1: IMPACT on CIFAR-10 and CIFAR-100. Performance on ResNet-50, 5000 Query Budget, and 5% Perturbation Area.*
>
> Note that due to the **lower resolution of CIFAR images (32 × 32)**, we adjusted the encoding to use a 1 × 1 tiling. This provides maximum granularity while keeping the search space manageable, demonstrating the **adaptability of our framework**. As the results in the table demonstrate, our method achieves near-perfect ASRs of **over 99%** in the untargeted setting on both CIFAR-10 and CIFAR-100, **surpassing** the already strong performance of Patch-RS. Notably, IMPACT consistently achieves a high attack success rate **above 90%**, even in the more challenging targeted attack setting across both CIFAR-10 and CIFAR-100. These results show that IMPACT's effectiveness is not confined to a specific data distribution like ImageNet. It generalizes well to datasets with varying characteristics.
>
> ### **W3. Code:**
>
> **We confirm that the complete, well-documented code will be released upon paper acceptance.** Our current supplementary code covers the **core components**.
>
> ### **Q1. Definition of $p_i$:**
>
> We thank the reviewer for pointing out this ambiguity. In our framework, the first phase of optimization utilizes a Differential Evolution (DE) algorithm. DE is a population-based algorithm. This means it maintains a **population**, which is a set of **candidate solutions**, and iteratively refines this population to find better solutions. We use $p_i$ to represent the $i$-th **individual** in the population, where $i \in [1, N]$ and $N$ denotes the population size. Each individual $p_i$ in the population represents a candidate solution that encodes all the necessary information to generate the patches. As we state in the paper, this information is divided into two components: the encoded mask $\mathbf{m}_i$ and the encoded content $\mathbf{r}_i$.
>
> To ensure this is clear in the final version of the paper, we will revise the paragraph starting at line 181. Furthermore, we will conduct a thorough review of the entire paper to identify and correct similar notational ambiguities.
>
> We appreciate the reviewer's careful attention to detail, which has helped us strengthen the clarity of our paper.
>
> ### **Q2. Analyze of the 4 × 4 Tiling Encoding:**
>
> We sincerely thank the reviewer for this insightful question, which highlights a critical **trade-off** in our design. We elaborate below on the rationale for adopting the 4 × 4 tiling scheme, highlighting trade-offs across **granularity**, **dimensionality**, and **optimization efficiency**:
>
> - **Granularity vs. Dimensionality**: The drop in ASR with smaller tile sizes is due to the **increased optimization difficulty under limited queries**. A 1 × 1 tiling provides the finest possible granularity, allowing precise per-pixel patch shaping. However, this results in a prohibitively large binary vector (224 × 224 = **50,176 dimensions**), which severely hampers the search efficiency of population-based optimization methods under black-box query constraints. Conversely, coarse tiling (e.g., 8 × 8 or larger) significantly **reduces dimensionality but lacks flexibility** in representing complex patch contents or irregular patch shapes.
>
> - **Empirical Trade-Off Analysis**: We performed extensive ablation studies with multiple tile granularities and observed that **4 × 4 tiling offers a favorable balance**: it maintains enough expressiveness to represent irregular patches while compressing the search space to a tractable size. This enables effective convergence within **limited query budgets**.
>
> - **Compatibility with Random Aggregation**: The 4 × 4 tiling grid also integrates well with our random aggregation algorithm. **Smaller tiles would create a more fragmented starting point, complicating the clustering process, while larger tiles would limit the shape's expressiveness.** The 4 × 4 tiling enables contiguous, locally coherent regions to **form effectively**, enhancing the real-world printability of generated patches.
>
> We will incorporate this expanded justification into Section 4.3 and include supporting ablation results. We thank the reviewer again for prompting this important clarification.
>
> ### **Q3. Performance in the Physical World:**
>
> We sincerely thank the reviewer for this important question. Demonstrating the physical-world applicability of our method is indeed a critical part of our evaluation.
>
> To validate the physical-world effectiveness of our method, we conducted real-world experiments following Wang et al. [4]. Specifically, we randomly selected 100 images with adversarial patches and printed each on a 10cm × 10cm white paper. Then, using an iPhone 15, we photographed each printed image at various distances and angles. All photos were then resized to 224 × 224 pixels for ResNet50 model input. Table 2 reports the resulting ASR across the different shooting distances and viewing angles. Note that all these results are for **targeted attacks**. This means the goal was not simply to cause a misclassification, but to force the model to predict a specific, pre-determined incorrect class.
>
> |Model|Digital ASR|Angle|ASR at 10cm|ASR at 15cm|ASR at 20cm|
> |-|-|-|-|-|-|
> |ResNet50|53.0|0°|41.0|37.0|32.0|
> |||15°|35.0|31.0|26.0|
> |||30°|28.0|25.0|21.0|
>
> *Table 2: Physical-world performance of IMPACT under different conditions. The targeted attack success rates ASR (%) is reported across varying distances and viewing angles for ResNet50 model. The "Digital ASR" indicates the attack success rate of the same 100 selected images in the digital domain.*
>
> These results confirm that our irregular patches **remain effective when physically deployed**. Even under the challenging condition (20cm distance and a 30° viewing angle), IMPACT still achieves a 21.0\% ASR. It is worth noting that even a strong baseline like Patch-RS only achieves a 20.0% ASR in the digital-world targeted attack setting (Table 1 in our main paper). In the physical world, varying distances and angles inevitably reduce the patch's detail in the captured image. Consequently, a degradation in performance when moving from the digital domain to the physical world is an expected phenomenon. **Despite these challenges, our method demonstrates resilience.** This validates the practical applicability and physical feasibility of IMPACT.
>
> In the final version of the paper, we will include the analysis of these real-world experiments. All physical-world images will be released upon acceptance of the paper.
>
> **We trust that our detailed responses have resolved your concerns. We would be sincerely grateful if you could kindly reconsider your assessment in light of these clarifications. Please feel free to let us know if there are any further questions. We would be happy to provide additional responses.**
>
> **Reference:**
>
> [1] Croce et al. Sparse-RS: A Versatile Framework for Query-Efficient Sparse Black-Box Adversarial Attacks. AAAI 2022
>
> [2] Chen et al. Query-Efficient Decision-Based Black-Box Patch Attack. TIFS 2023
>
> [3] Rao et al. Adversarial Training Against Location-Optimized Adversarial Patches. ECCV 2020
>
> [4] Wang et al. Breaking Barriers in Physical-World Adversarial Examples: Improving Robustness and Transferability via Robust Feature. AAAI 2025

---

> > ### Comment · Reviewer_d9ew · 2025-08-05
> >
> > I appreciate the authors' efforts during the rebuttal period. Their explanations regarding the dataset, $p_i$, and tiling encoding have addressed my concerns. I am willing to raise my score from 2 to 3.

---

> ### Author Response · Authors · 2025-08-06
>
> We sincerely thank you for your positive reassessment and are delighted that our responses have addressed your concerns regarding the dataset, $p_i$, and tiling encoding.
>
> In addition to those points, we also hope our responses regarding **the contribution clarification** and **the physical-world performance** have also been helpful:
>
> - **Contribution**:
>
> While our encoding uses two vectors, the optimization process in the **core first phase of our framework** is fundamentally a joint optimization. In this phase, a candidate solution, represented by the tuple $(\mathbf{m},\mathbf{r})$, is treated as **a single indivisible entity**. That is, the decision variables of $\mathbf{m}$ and $\mathbf{r}$ are represented by an individual and optimized simultaneously by a differential evolution algorithm, such that the method actually unifies the elements (shape, location, number, and content) into a common search space.
>
>
> The key advantage of this joint optimization is its ability to **discover synergistic combinations** amongst these factors. This is a crucial distinction compared to prior methods, as their sequential optimization approaches tend to get trapped in local optima and miss these globally optimal solutions, thereby being challenging to achieve a high ASR. As shown in Table 2 in our main paper, this joint optimization phase (Phase 1) alone is highly effective, achieving an ASR of 90.2%, which is already higher than competing SOTA methods.
>
> The second phase is a fine-tuning step. It operates under the condition where the patch mask is already fixed by the superior solution found in Phase 1. The purpose of the second phase is to perform a pixel-level refinement of the patch **content** (decision variable $\mathbf{r}$) to further boost the final ASR. Therefore, the core of our framework (Phase 1) is a joint optimization process, and it is the primary driver of our method's success.
>
> To the best of our knowledge, our work is the first to study the joint optimization of shape, location, number, and content for adversarial patch attacks. To successfully tackle this problem, we design the IMPACT framework with a dimensionality reduction encoding scheme and a new aggregation algorithm that merges scattered patch elements into locally coherent shapes.
>
> - **Physical-World Performance**:
>
> We provided real-world print-and-capture experiments to demonstrate the practical applicability of our method.
>
> **We hope these clarifications address all your concerns. If so, we would be sincerely grateful if you might consider whether this paper could merit a more positive rating. Thank you again for your insightful feedback, and we remain available for any further questions.**

---

### Note · Authors · 2025-08-12

Dear Area Chairs and Reviewers,

As the discussion phase concludes, we sincerely thank you for your time, thoughtful evaluations, and constructive feedback. We are encouraged by the positive engagement during the rebuttal period and believe that all concerns have been satisfactorily addressed.

We would like to highlight several key improvements made in direct response to reviewer feedback:

**1. Expanded Empirical Validation**

- **Physical-World Experiments**: We conducted real-world experiments, confirming IMPACT's effectiveness across distances and angles, demonstrating its practical applicability.

- **Cross-Dataset Generalization**: We added new experiments on CIFAR-10 and CIFAR-100, showing that IMPACT generalizes effectively beyond ImageNet, achieving over 90% ASR even in the challenging targeted attack setting.

- **Runtime Analysis**: We provided a runtime analysis showing that IMPACT remains computationally practical, while delivering higher ASR and requiring fewer queries, which is a dominant cost factor in black-box attacks.

**2. Clarifications and Enhancements**

- **Contribution**: We strengthened the explanation of our joint optimization framework and clearly articulated its advantages over sequential or factor-isolated methods, highlighting its ability to capture complex interdependencies among all patch factors.

- **Tiling Encoding**: We analyzed the 4×4 tiling design, explaining its optimal balance between solution expressiveness and search space complexity, supported by empirical results.

- **Notation**: We addressed ambiguities in notation and algorithm descriptions to enhance the paper's readability.

**3. Future Research Directions**

- We expanded our discussion to outline promising future extensions of IMPACT, including evaluation against diffusion-based defenses, adaptation to decision-based attacks, and potential theoretical analysis.

With these substantial additions and clarifications, we are confident that our responses fully addressed all reviewer concerns. All new experiments, analyses, and explanations will be incorporated into the final version of the paper. In addition, we also improved the section flow and terminology for better clarity. We deeply appreciate your consideration and hope our efforts will earn your support in the final decision.

Finally, we express our heartfelt thanks again to area chairs and all reviewers for your time, effort, and constructive feedback.

Sincerely,

The Authors of Submission 21646

---

### Decision · Program_Chairs · 2025-09-17

**Decision:**

Accept (poster)

**Comment:**

IMPACT delivers an advance​ in adversarial patch attacks by unifying previously disjointed optimization tasks into a cohesive, efficient framework. Its encoding scheme and two-phase optimization resolve core limitations of prior work, achieving ​superior attack success​ with ​greater practicality. The authors' rigorous validation—spanning digital, physical, and cross-dataset settings—and responsive engagement with reviewers solidify this contribution. Given the technical novelty, empirical strength, and resolved reviewer consensus, I endorse ​acceptance.